# Little red dots as young supermassive black holes in dense ionized cocoons

V. Rusakov[1,2,3 ✉], D. Watson[2,3], G. P. Nikopoulos[2,3], G. Brammer[2,3], R. Gottumukkala[2,3], T. Harvey[1], K. E. Heintz[2,3,4], R. Damgaard[2,3], S. A. Sim[2,3,5], A. Sneppen[2,3], A. P. Vijayan[6], N. Adams[1], D. Austin[1], C. J. Conselice[1], C. M. Goolsby[1], S. Toft[2,3] & J. Witstok[2,3]

The James Webb Space Telescope (JWST) has uncovered many compact galaxies at high redshift with broad hydrogen and helium lines, including the enigmatic population of little red dots (LRDs)[1,2]. The nature of these galaxies is debated and is attributed to supermassive black holes (SMBHs)[3,4] or intense star formation[5]. They exhibit unusual properties for SMBHs, such as black holes that are overmassive for their host galaxies[4] and extremely weak X-ray[6–10] and radio[6,11–13] emission. Here we show that in most objects studied with the highest-quality JWST spectra, the lines are broadened by electron scattering with a narrow intrinsic core. The data require very high electron column densities and compact sizes (light days), which, when coupled with their high luminosities, can be explained only by SMBH accretion. The narrow intrinsic line cores imply black hole masses of $10^{5-7}M_\odot$, two orders of magnitude lower than previous estimates. These are the lowest mass black holes known at high redshift, to our knowledge, and suggest a population of young SMBHs. They are enshrouded in a dense cocoon of ionized gas producing broad lines from which they are accreting close to the Eddington limit, with very mild neutral outflows. Reprocessed nebular emission from this cocoon dominates the optical spectrum, explaining most LRD spectral characteristics, including the weak radio and X-ray emission[14,15].

Two early clues suggested that the compact galaxies may be affected by Compton-thick ionized gas: Balmer absorption features are often observed in the broad lines[2,16,17], and if they are active galactic nuclei (AGN), their X-rays might be suppressed by photoelectric absorption in this gas in the broad-line region (BLR) near the SMBH[10]. These ideas inspired us to investigate the Hα line profiles to look for electron-scattering signatures produced by the large ionized gas column densities.

To test different broad line shapes, we constructed a sample of all broad-line galaxies (Hα ≳ 1,000 km s⁻¹) with high signal-to-noise ratio (SNR) JWST/NIRSpec medium-resolution (R = 1,000) spectra in the DAWN JWST Archive (DJA)[18,19]. Details of the sample and selection criteria are presented in the Methods and Extended Data Table 1. Our search yielded 12 objects at z = 3.4–6.7 and 18 additional objects for a combined 'stacked' spectrum at z = 2.32–6.76. High-resolution (R = 2,700) data were used where available (objects A and C), which gave consistent results.

Our object selection is strongly linked to LRDs, many of which show broad lines. LRDs are spatially compact with characteristic v-shaped optical-UV spectra[1], because of a change of slope close to the Balmer limit wavelength, about 365 nm (refs. 20,21). Despite our selection being based only on Hα linewidth and SNR criteria, our systems are all spatially very compact (Fig. 1) and often have a clear slope change at the Balmer limit in the rest frame spectra (Extended Data Figs. 1 and 2).

In agreement with previous findings[22,23], most higher-ionization lines often associated with classical AGN, such as He II, C IV, N V and [Ne V], are absent in all 5 out of the 12 high-SNR sources that have rest-UV coverage, whereas the highest-ionization lines we detect with SNR > 5 are [Ne III] λ3,867, 3,967, in three out of seven sources with the spectral coverage, with equivalent widths of 25–50 Å.

Electron scattering in a dense ionized gas produces lines with exponential profiles[24–26], whereas Gaussian or centrally broad profiles are expected from Doppler broadening[27]. We, therefore, compare two basic models for the broad Hα component: a Gaussian and a double-sided exponential. Apart from the broad component and a local continuum in the Hα region, we model the narrow emission lines of Hα and [N II]λλ6,549, 6,585. Our model also includes narrow absorption features (object E) or P Cygni absorption and emission features (objects A and D) that appear in one third of the objects here (Fig. 3). Following our model comparison, the exponential model is superior in almost every case (Extended Data Fig. 3 and Extended Data Table 2), whereas the best Gaussian fits leave systematic residuals with a characteristic W shape around the central region (Fig. 2). The significance of the fit improvement increases with the SNR of the spectra (Extended Data Fig. 4), indicating that the exponential model is generally better, even where the SNR is not high enough to discriminate. The lines are symmetric and trace a straight line on a semi-logarithmic plot over several orders of magnitude, consistent with moderate

[1]Jodrell Bank Centre for Astrophysics, University of Manchester, Manchester, UK. [2]Cosmic Dawn Center (DAWN), Copenhagen, Denmark. [3]Niels Bohr Institute, University of Copenhagen, Copenhagen, Denmark. [4]Department of Astronomy, University of Geneva, Versoix, Switzerland. [5]Astrophysics Research Centre, The Queen's University of Belfast, Belfast, UK. [6]Astronomy Centre, University of Sussex, Brighton, UK. ✉e-mail: rusakov124@gmail.com

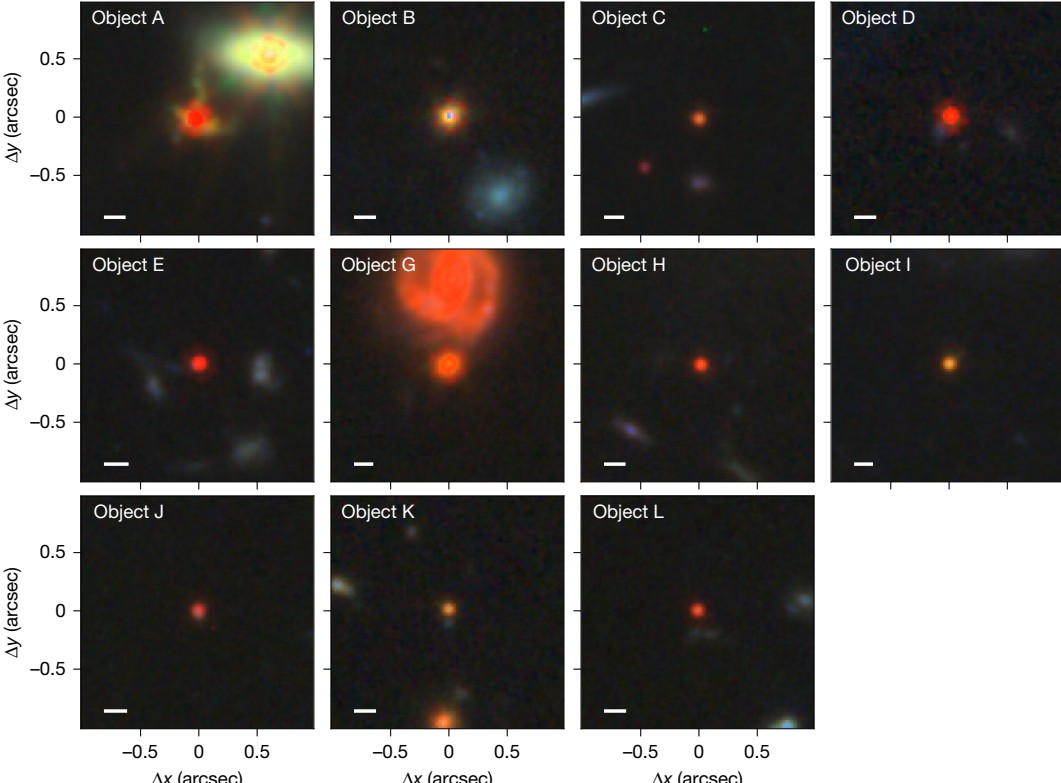

**Fig. 1 | JWST/NIRCam images of the sample objects.** Despite being selected based on the presence of broad Hα components, all objects appear to be point-like or very compact, occasionally with some nebulosity. Rest frame optical colours are mostly red. These properties are similar to LRDs (Extended Data Figs. 1 and 2). Object A has an unusual extended cross-like structure. We note that this object also shows rotation in the 2D high-resolution spectra of the narrow nebular lines. Object F was not observed by NIRCam. The RGB colouring is scaled using a wavelength-dependent power-law (with exponent of 2) based on the fluxes in the F150W (or F200W), F277W and F444W bands. The physical scale (1 kpc) is indicated as a white bar in the lower left corners of the images. Each snapshot is 2 arcseconds wide.

optical depth electron-scattering[25]. These results indicate that the primary line-broadening mechanism is electron scattering through a Compton-thick medium and not primarily bulk Doppler motions, significantly reducing the inferred SMBH masses. The symmetry of the lines suggests that any net outflow of the scattering medium is less than a few hundred km s$^{-1}$ (ref. 26). We also consider Lorentzian profiles (due, for example, to Raman scattering or turbulence[27]) and multiple Gaussians (due to complex kinematics or orientation effects[28]) and find the exponential model to be statistically better in most cases (Methods and Extended Data Table 2).

After establishing that the basic broad line shape is dominated by electron-scattering effects, we allow for an intrinsic Doppler line core. For our fiducial model, we, therefore, use a Gaussian convolved with an exponential, with the widths free to vary, to measure the intrinsic width of the Doppler lines. Fits for all objects are shown in Fig. 3. In most cases, the intrinsic Gaussian line width is small, with average velocities of approximately 300 km s$^{-1}$ for 9 of the 12 objects in the sample. In objects B and D, the core width is 1,500–2,000 km s$^{-1}$, although their total line widths are still dominated by broadening from electron scattering. Finally, only object G has a dominant Gaussian width of 2,000 km s$^{-1}$.

The characteristic properties of the electron-scattering gas can be inferred from the line wings. First, we estimate the free-electron column density, $N_e$, from the width of the exponential component[25,26]. We use our own Monte Carlo electron-scattering code with a simple shell geometry to calculate the relationship between the electron-scattering optical depth, $\tau_e$, the temperature, and the characteristic width of the exponential (Methods). We find $\tau_e = 0.5$–2.8, implying that the column density is approximately distributed as $N_e \simeq 0.7$–$4.2 \times 10^{24}$ cm$^{-2}$ for the sample, assuming an electron temperature between 10,000 K and 30,000 K (Extended Data Table 1 and Extended Data Fig. 7). The absence of strong broadening of the [O III] $\lambda\lambda$4,959, 5,007 doublet, whereas the Hβ line next to it is broadened, suggests that the electron volume density, $n_e$, in the scattering region, must be at least several times the critical density for these lines. The characteristic spherical size of the scattering region, $R_c$, can be inferred by taking the ratio of the column and volume densities in a simple constant density sphere, that is, $R_c \sim 3N_e/n_e$. Assuming $\log[n_e(\text{cm}^{-3})] \gtrsim 6.5$, we find sizes of at most a hundred light days. With higher densities[14,29] (that is, $\log[n_e(\text{cm}^{-3})] \gtrsim 8$), the sizes would be smaller (about a few light days).

The dense Compton-thick gas inferred from the line wings offers a natural explanation for the Balmer absorption lines observed frequently at high redshift[2,29]. Given the extreme column and volume densities of the ionized gas inferred here, the fraction of the gas at the $n = 2$ level should provide sufficient optical depth to produce these features. We model the features with P Cygni profiles for objects A and D, as they fit the characteristic sharp transition between the emission peak and the blueshifted absorption[30] much better than pure absorption. The properties of the absorption features indicate a largely spherical gas distribution, with mild outflow velocities below a few hundred km s$^{-1}$. In cases of high optical depth and lower velocities, radiative transfer effects may induce a complex self-absorption feature in the line centre, which may be filled by narrow emission lines from the rest of the galaxy, making it difficult to assess how common such Balmer absorption is.

We now turn to the origin of the lines in these systems. The line cores in most of our sample could be explained by very mild outflows, consistent with the photospheric velocities we find for the P Cygni lines in objects A and D (about 200–300 km s$^{-1}$). This mild outflow could be due to feedback from a burst of star formation. However, the brighter

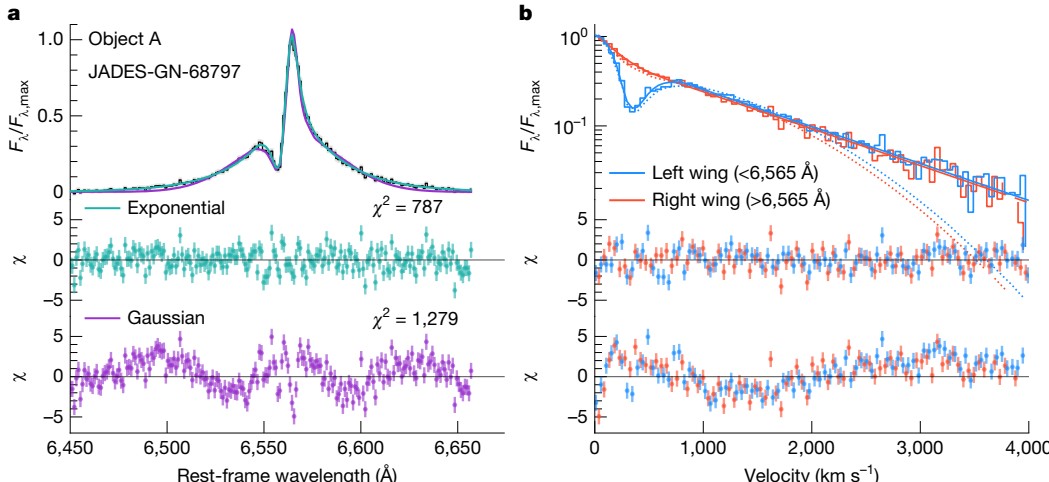

**Fig. 2 | The Hα line profile for JADES-GN-68797 (object A).** The models in the figure compare the fiducial best-fit model that includes a scattered exponential component with an identical model that instead includes a double-Gaussian component. Object A has the highest SNR(Hα) in our sample and was observed with the high-resolution grating on JWST/NIRSpec. The absorption feature is modelled with a P Cygni component. Left, Gaussian and exponential-component models compared in linear flux density space. The residuals of Gaussian and exponential models are shown in terms of the number of standard deviations. Right, Hα line profile plotted in semi-logarithmic space, reflected about the line centre, showing the linearity of the line profile in this space. The wings of the lines are also symmetric. The exponential line shape is significantly preferred by the data both around the line core and the tails, whereas the Gaussian model falls below the line core and the tails.

systems in our sample have ionizing luminosities of $\gtrsim 10^{45}$ erg s$^{-1}$, based on observed Hα luminosities around $10^{43}$ erg s$^{-1}$ (ref. 31), which is nearly four orders of magnitude higher than in those star clusters with typical luminosity densities of about $10^{41}$ erg s$^{-1}$ per pc$^2$ or per pc$^3$. Distributing the inferred luminosity across multiple sources over a larger volume (for example, many accreting extremely massive stars[32]) does not resolve this problem because the inferred gas masses (using $N_e \sim 10^{24}$ cm$^{-2}$ and $n_e \lesssim 10^8$ cm$^{-3}$) require ionizing luminosities $\gtrsim 10^9 L_\odot$ to keep each of them ionized. Only AGN accretion can realistically power such large ionizing luminosities in regions under a few hundred light days. This upper limit is consistent with the prediction from AGN radius–luminosity relation[33] of approximately 5 light days for a Hα luminosity of $10^{43}$ erg s$^{-1}$.

Our analysis eases or resolves some of the challenges faced by the AGN interpretation. First, removing the effects of electron scattering leaves intrinsic line cores 10 times narrower than those obtained from a simple Gaussian fit to the full line profile (the deconvolved black Gaussians in Fig. 3). Assuming these intrinsic line cores are entirely due to orbital motion around the SMBH, the new inferred black hole masses are, therefore, lower by about a factor of a hundred (Extended Data Table 1). Our estimates of the maximum SMBH masses are consistent with the SMBH–host galaxy scaling relation found at lower redshifts[34] (Fig. 4), alleviating much of the black hole mass and early growth problems, and the number density problem. Second, we infer a high column density of ionized gas, which should suppress soft X-ray emission by photoelectric absorption, even at low metallicities, because of He absorption[35]. However, the column densities we infer here will attenuate the hard X-rays only by a factor of a few (Methods)—less than the $\gtrsim 1$ dex required to satisfy the observations[6–10]. Therefore, more importantly, steep hard X-ray spectral slopes and/or power-law cut-offs at relatively low spectral energies are also required to suppress the X-rays. These steep hard X-ray slopes are found in high-accretion rate AGN, such as narrow-line Seyfert 1 (NLS1) objects[36], possibly because of enhanced Compton cooling of the corona by strong soft X-ray emission from a bright disk[37]. Finally, the high-density gas cocoon may help to explain the radio non-detections[29] by free-free absorption[38] and by suppressing jet formation by baryon loading.

The high-density gas cocoon reprocesses essentially all of the Lyman continuum radiation from the central source, resulting in extremely high Lyman optical depths that cause the self-reversal seen as absorption in the line centres of some of the Balmer lines. The bulk of the ionizing flux from the AGN is, therefore, emitted by recombination, implying that nebular gas emission must be the main component of the optical and NIR spectra of these sources, giving rise to Balmer, Paschen and He I emission lines and continua and the two-photon continua that dictate the spectral shape[14,39]. We also note that our broad lines are highly symmetric in all examples (Extended Data Fig. 3), unlike in type 1 quasars that demonstrate Balmer line asymmetries in up to a quarter of cases[40], and inferred electron-scattering optical depths are all $\tau_e \sim 1$ (Extended Data Fig. 7). This suggests three things: (1) there may be a population that is even more heavily obscured, suppressing line emission due to self-absorption and giving rise to Balmer breaks instead of jumps[15,41,42]; (2) the gas distribution is close to spherical, without a large opening angle, as otherwise the light will preferentially escape along the lowest column density sightlines with less scattering. The most obvious interpretation of these facts is that the scattering medium and a BLR are more or less the same (quasi-spherical) material that emits and scatters the broad lines in its inner regions. With a decreasing radial density, the inner regions produce far more flux than the outer regions, which provide most of the scattering opacity. This is feasible without invoking an extreme narrow-to-broadened line ratio as recently suggested[43]. The lack of low-column density sightlines may be explained with low metallicity of high-redshift AGN[4], which may hinder efficient cooling and decrease clumping in the BLR leading to a smoother ionized gas cocoon; (3) the recombination physics may deviate from the standard case B scenario because of very high Lyman optical depths[44], so that high Hα/Hβ ratios do not necessarily imply dust extinction[45,46].

Another conundrum of these sources was their low inferred accretion rate[17]. Early stages of black hole growth require high accretion rates over long times to grow rapidly. The lower black hole masses we infer here solve this puzzle. We calculate a mean Eddington ratio around unity for our sources A–L, excluding objects B and G, assuming the Hα line is a few per cent of the bolometric luminosity. As our sample is likely to be biased in favour of more luminous systems, the substantial part of the population may be in the intermediate-mass black hole regime with lower accretion rates, and/or greater dust extinction.

All of these point to a solution to the riddle of LRDs/compact broad-line objects of JWST. They are intrinsically narrow-line AGN, that is, young, low-mass ($10^5$–$10^7 M_\odot$), Eddington-accreting SMBHs, buried

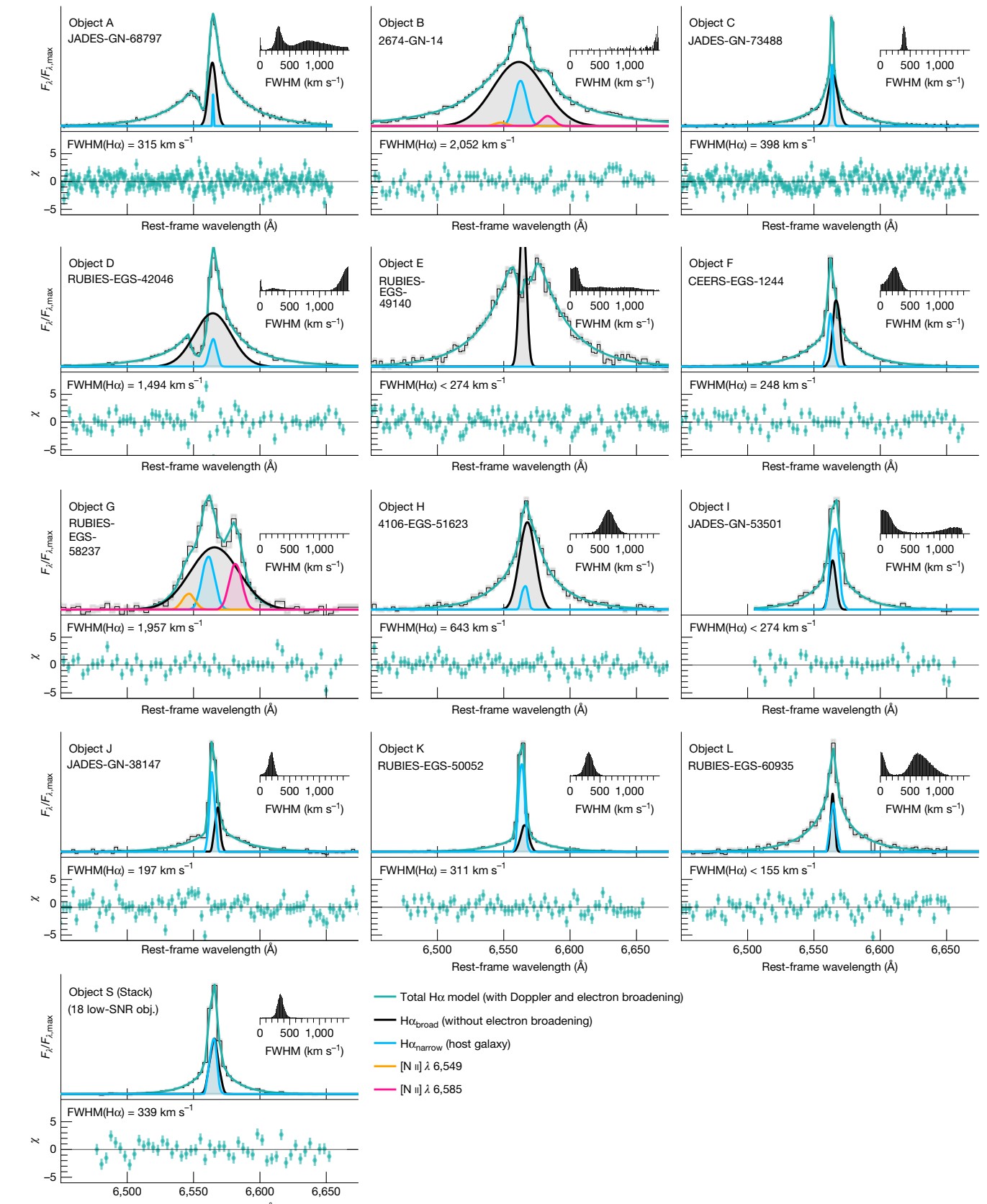

**Fig. 3 | Hα line profiles for the full sample fitted with the fiducial scattering model.** The total model (cyan line) is the broad scattered and non-scattered components of the intrinsic Gaussian line (black line and grey fill), and a narrow Gaussian Hα component from the host galaxy (blue line), as shown in the legend. The residuals of the best-fit total model are shown in cyan below each spectral line in terms of the number of standard deviations. In objects A, D and E, a P Cygni profile or a Gaussian absorption feature are included. The insets show the FWHM posterior of the Doppler component of the broad Hα (limited to <1,500 km s⁻¹ for clarity). In all cases except for objects B and G, which are complex and may not be well-modelled by our simple approach, the widths of the broad Hα lines are dominated by the electron-scattering mechanism (1,000–2,000 km s⁻¹), whereas the Doppler motions are either unresolved or on the order of several hundred km s⁻¹.

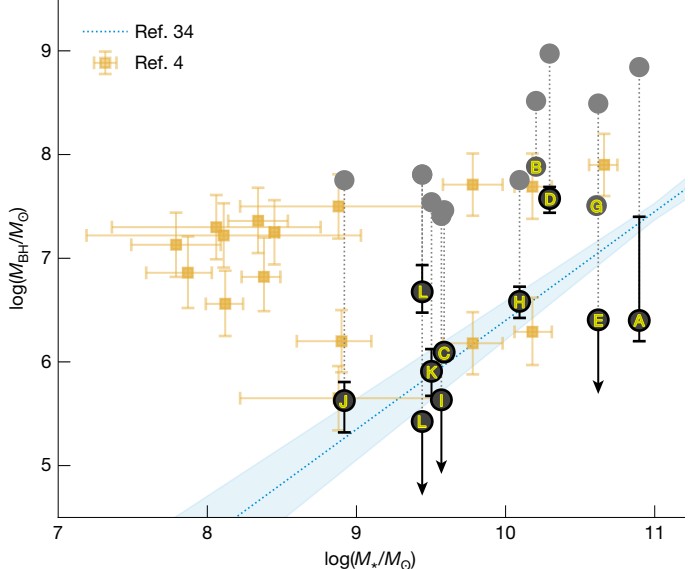

**Fig. 4 | SMBH mass compared with the stellar mass of the host galaxies.** The SMBH masses are determined from the Doppler components of our fiducial model (letter-labelled points) using a virial relation[49]. Their uncertainties represent the 16th and 84th percentiles of the posterior distributions, while the upper limits are defined as $2\sigma$ values. Objects B and G may have complex absorption features that could substantially change their $M_{BH}$—they are therefore plotted only in grey. The posterior of $FWHM_{Doppler}$ for object L is strongly bimodal (Extended Data Table 3), so $\log(M_{BH})$ is split into two points. Previous estimates from the literature are also shown (yellow points), most of which are included in our low-SNR stack, which has median inferred $\log(M_{BH}) \lesssim 5.6$. We also show the black hole masses inferred with a Gaussian model without electron scattering (grey points) to illustrate the effect of the scattering model. Removal of the electron-scattering line broadening reduces the inferred SMBH masses by about two orders of magnitude and makes the masses consistent with the observed relation of AGN at lower redshift[34]. However, the stellar mass $M_\star$ inferred from SED fitting is an upper limit, since a large fraction of the emission in the optical is expected to be reprocessed from the AGN, rather than from stars (consistently with ALMA non-detections of molecular gas and dynamical mass estimates[50]). $M_{BH}$ may be overestimated (by <0.5 dex) due to a possibly higher BLR coverage fraction (Methods). Moreover, $M_{BH}$ may be underestimated if there is a substantial dust extinction in the BLR (for example, an $A_V = 5$ would increase $M_{BH}$ by about 0.75 dex).

in a thick cocoon of gas, presumably related to their youth[47]. Their high accretion rates produce copious UV emission that ionizes their gas cocoon and at the same time efficiently cools and weakens the corona, suppressing their hard X-rays[48]. The ionized gas hinders the escape of radio and X-rays, while reprocessing almost all the Lyman radiation into nebular optical emission, producing the broadened Balmer lines and continuum breaks that characterize the classic v-shaped spectrum[21]. This low-metallicity[4] ionized gas cocoon does not clump efficiently and has a smooth distribution leaving few optically thin sightlines. The distribution of electron column densities and the lack of a similar population of low-mass, high-accretion AGN hints that we may be observing the bulk growth of SMBHs in the phase when they are surrounded by a quasi-spherical, dense gas shell, before metallicity effects and efficient winds have cleared their polar regions and opened up the cocoon.

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

## Methods

In this paper, uncertainties are given as 1$\sigma$ or 68% confidence intervals. Upper limits are indicated at the 2$\sigma$ level unless otherwise stated. We adopt cosmological parameters measured in ref. 51, that is, a $\Lambda$ cold dark matter ($\Lambda$CDM) model with total matter density in units of the critical density, $\Omega_m = 0.310$, and Hubble constant, $H_0 = 67.7$ km s$^{-1}$ Mpc$^{-1}$.

### Spectroscopic sample

This study makes use of the public JWST data collected as part of several observational programmes with the NIRSpec spectrograph[52] with PIDs: 1345 (CEERS)[53], 1181 (JADES)[54], 1210 (JADES)[55,56], 2674 (PI A. Haro)[57], 3215 (JADES Origins Field)[58], 4106 (PI E. Nelson)[59], 4233 (RUBIES)[18], 2565 (PI K. Glazebrook)[60], 2750 (PI A. Haro)[61] and 6541 (PI E. Egami)[62]. These observations have been uniformly reduced and published as part of the DAWN JWST Archive (https://dawn-cph.github.io/dja) (DJA)[18,19]. Using the v.3 reductions in DJA, we selected all galaxies observed in the medium-resolution grating spectra with a broad H$\alpha$ component and a spectroscopic redshift produced by msaexp (https://github.com/gBrammer/msaexp)[63]. To this, we added JWST broad-line objects reported in the literature with publicly available data and processed in DJA. We selected objects with a full width at half-maximum (FWHM) linewidth greater than about 1,000 km s$^{-1}$ from the objects in the archive. We then selected spectra with high SNR (median SNR >5 per 10 Å for the continuum-subtracted region ±2,000 km s$^{-1}$ around the H$\alpha$ line) and also included objects for the stacked spectrum using broad-line objects with lower SNR (5 > SNR/10 Å > 1) to ensure that we are not biased by our SNR selection.

We note that our sample spans a range of colours when using the existing selection criteria (Extended Data Fig. 2). Although only three objects of 12 are classified as LRDs (AEH), more objects have a similar inflection point around the Balmer series limit (CDGH, as seen in the PRISM spectra in Extended Data Fig. 1) which is not picked up by the selection criteria due to redshift effects and the contribution of the strong optical emission lines making the colour gradient flatter. However, some objects are bluer than photometrically selected LRDs (CFI). Therefore, although most of our sample has classical LRD-like features, some objects probably have a wider range of properties, possibly produced by differing extinctions or contributions from the host galaxy. This difference can possibly be explained, in part, by the incompleteness of current photometric selection criteria in bluer F277W–F444W colours and fainter rest-UV magnitudes[64]. Despite this range of colours, the presence and magnitude of electron scattering by the ionized gas cocoons does not depend on the location in this colour space (as suggested by the optical depth in Extended Data Table 1 or exponential width in Extended Data Table 3).

### Emission line models

All best-fit results in this paper were produced using the Monte Carlo Markov Chain (MCMC) NUTS sampler as part of the package PyMC v. 5.17.0 (ref. 65), except objects A and D, which were fitted using the Ensemble sampler emcee v. 3 (ref. 66) (due to an incompatibility of the P Cygni model used here with the tensor formalism in PyMC). We sampled the posterior distributions with 4$k$ walkers (where $k$ is the number of free parameters) and 10$^5$ samples per walker. We use the mode values of the posterior parameter distributions as the best values and the 68% highest-density interval as the range of uncertainty. Finally, we find that the resolution of NIRSpec gratings is higher than the nominal value. Using the resolved widths of narrow [O III] lines in the high-resolution G235H grating of object A, we estimate that the medium-resolution grating G235M has about 1.7 times higher $R$ than the nominal value. This scaling factor on $R$ has been assumed for G395M and also for the G395H grating, which agrees with the results of forward modelling of the NIRSpec instrument response for point sources[67].

In modelling the broad H$\alpha$ profile, we assumed a broadening mechanism: either a Doppler velocity broadening or a Compton scattering broadening. The former is modelled using a Gaussian function $f(\lambda; A, \mu, \sigma) = A\exp((\lambda - \mu)^2/(2\sigma^2))$ with amplitude $A$, line centre $\mu$ and velocity dispersion $\sigma$. For the Compton-scattered profile we use a symmetric exponential $g(\lambda; B, \lambda_0, W) = Be^{-|\lambda - \lambda_0|/W}$ with amplitude $B$, line centre $\lambda_0$, and e-folding scale $W$.

First, we tested both broadening mechanisms by fitting two sets of models. Extended Data Fig. 3 shows the comparison between the broadening models. To fit the data reasonably well, the models also included (narrow) Gaussians for the host galaxy H$\alpha$ and [N II] doublet with fixed centroids and velocity dispersions tied to the same value and limited to <1,000 km s$^{-1}$. The ratio of the two [N II] lines is set to 0.33 (ref. 68). In some cases, an additional Gaussian absorption component (object E) or a P Cygni profile (objects A and D) are required to accurately model the broad H$\alpha$ component. We note here that the [N II] lines are only required in the fits in objects B and G, which fitted better than redshifted absorption. Whether this is an artefact related to the complex spectral shape due to possible self-reversal in the line, or whether [N II] really is observed in these cases, is unclear and would require higher resolution spectra and more sophisticated models to establish. Finally, wavelength regions around the emission lines [O I] $\lambda$6,302, He I $\lambda$6,678 and [S II] $\lambda\lambda$6,717, 6,731 are excluded from the fit.

Although most H$\alpha$ lines in the sample are predominantly exponential with very high statistical significance (Extended Data Fig. 3), to reconstruct the intrinsic Doppler widths, we model the lines with a Gaussian convolved with an exponential, instead of a pure exponential. These models are convolved with the instrumental resolution of the relevant gratings (which were taken from the JWST JDox website for NIRSpec) at the H$\alpha$ peak (we assume the actual resolution is about 1.7 times better than the nominal; see the description above). To alleviate the complexity of some of our models and more accurately constrain the narrow H$\alpha$ components, we use the velocity widths of the optical [O III] lines as a Gaussian prior on H$\alpha$ and [N II] widths, in which relevant spectral coverage is available. The best-fit profiles and intrinsic Doppler components with their widths are presented in Fig. 3 and Extended Data Table 3.

We also test whether, for example, gas turbulence or Raman scattering could be responsible for line broadening by comparing a basic Lorentzian[27,69] and an exponential line shapes. The former is defined as a symmetric profile with FWHM 2$\gamma$ centred at $\lambda_0$: $h(\lambda; C, \lambda_0, \gamma) = C\frac{\gamma}{(\lambda - \lambda_0)^2 - \gamma^2}$. The exponential is a significantly better fit in most objects or an equivalent fit in objects H and L (Extended Data Table 2 and Extended Data Fig. 5). This indicates that any potential contribution from turbulence broadening is not significant, we do not assume a more physically motivated profile of a Gaussian convolved with Lorentzian (that is, a Voigt profile). Another reason to exclude this model is that the ratios of the line widths between H$\alpha$, H$\beta$ and Pa$\beta$, for example, are expected to differ by about a factor of 2–3 in velocity[27], something which is not generally observed in these types of objects. Furthermore, turbulent broadening may sometimes result in enhanced red-wing profiles[69], whereas all objects here have symmetric H$\alpha$ wings (Extended Data Fig. 3). However, although it has been argued that the lines cannot be Lorentzian on this basis[29], optical depth effects, which would be quite different for the different lines, could affect the relative line widths and more careful non-LTE radiative transfer analysis would help explain this issue.

Finally, we investigate the possibility that double-Gaussians are required to model the broad-line components, as has been reported in studies of similar JWST objects[4,43,70]. Multiple Gaussians may arise due to outflows, orientation effects of accretion disks[28,71], biconical geometries[72] or as a combination of motion of the outer accretion disk and the outer BLR[73]. By fitting the double-Gaussian model in our sample (apart from a narrow component), we find that both broad Gaussian components are always at precisely the same centroid position, which reflects the line symmetry and rules out the possibility of outflows, orientation

effects or biconical BLR typically producing double-peaked or flat-top line profiles[28,71,72]. It is also unlikely that binary AGN are responsible for producing these profiles in all objects in the sample (which is suggested for similar systems with strongly non-Gaussian broad lines in ref. 4) as the occurrence rate of binary AGN is predicted to be only a few per cent at intermediate redshifts in simulations[74] and line asymmetries would be expected in the default case. By contrast, symmetric profiles requiring fitting with two broad Gaussians might be explained with a stratified BLR[73] or (if the intrinsic profile is actually Lorentzian) a turbulent outer accretion disk and have been identified at $z \sim 2$ (ref. 75). We cannot perform an equal comparison here, as the latter study considers only single and binary Gaussian profiles of Balmer lines, but because their sample has luminosities roughly two orders of magnitude higher than ours, it is possible that the broadening in these systems is produced by turbulence rather than electron scattering. Therefore, we note that these objects represent an interesting sample for a future analysis with exponential line profiles.

Here, even a double-Gaussian model is statistically disfavoured for 8 out of 12 of our objects based on $\Delta$BIC > 10, as well as for the spectral stack. The $\Delta$BIC is greater at higher SNR, suggesting again that this is a real effect. In other cases, the double-Gaussian broad H$\alpha$ provides similar goodness of fit to the fiducial model, albeit with one more parameter, or fits better than the fiducial model by $\Delta$BIC $\approx$ 10 for objects B and J. Finally, the FWHM of both components has identical posterior distributions in 7 out of 13 cases implying they are degenerate and not physical, whereas in the other cases (objects A, D, E, H and S), the narrower component reaches a width 20–40% of that of the broader one, to better fit the core and the extended wings. As there are no clear inflection points in the overall shape of the broad lines in the latter examples, this suggests that the two broad-line components are probably mimicking the single exponential shape. Based on these properties, it seems unlikely that the double-Gaussian model is more physical than the exponential, and it is rejected based on its statistical performance. We show the comparison between the fit statistics ($\chi^2$ and BIC) for all tested models (Gaussian, double-Gaussian, exponential, Lorentzian and fiducial) in Extended Data Figs. 4 and 5 and Extended Data Table 2.

### Optical depth measurement

To estimate the approximate optical depth of the scatter from the exponential linewidth measurement ($W$), we use Monte Carlo simulations of electron scattering at 10,000 K and various optical depths in a spherical shell geometry. Specifically, we simulate Compton scattering in the low-energy limit ($h\nu \ll m_e c^2$) in a thin, uniform density/temperature spherical shell ($\Delta r/r = 0.1$) that is characterized by its radial optical depth to electron scattering. We assume photons enter the scattering region at normal incidence and follow their scattering until they emerge from the simulation domain. We simulate only the Compton scattering process in the domain (no true absorption or line scattering is included). The relation between scattering optical depth and the width of the scattered exponential scales almost linearly (Extended Data Fig. 6). We model the relationship as

$$W = a\tau + b,$$

where $a$ = 428 km s$^{-1}$ and $b$ = 370 km s$^{-1}$.

The relationship scales roughly as the square root of the temperature[25], so that at temperatures of 20,000 K the inferred optical depths would be 30% lower, for example.

### Spectral stacking

Individual noisy spectra are combined to obtain a median spectral stack with a greater SNR in the 4,000 km s$^{-1}$ region centred on the H$\alpha$ line. Initially, we fit each H$\alpha$ line profile with a basic exponential broad H$\alpha$ component with its amplitude $B$ and e-folding width $W$. Then we subtract the best-fit continuum around H$\alpha$ and normalize the individual widths of the line profiles to the median $W$ of the spectral stack. Similarly, individual exponential amplitudes $B$ are normalized to the median amplitude. Next, we resample the spectra to the same rest-frame wavelength grid oversampled by a factor of 10 with respect to the highest resolution H$\alpha$ spectra in the stack to avoid aliasing effects. Finally, the stacked spectrum is produced by taking the median of the stack and estimating uncertainties by drawing 10,000 times from Gaussian uncertainties of individual spectra. In the end, this stacked spectrum is resampled to the wavelength grid with the resolution equal to the median of the spectral stack resolutions.

### Reliability of the black hole mass estimation

As the properties of AGN in this work are distinct from typical type I AGN, our masses (reported in Fig. 4 and Extended Data Table 1) are extrapolations of those typical black hole mass estimates. In particular, the estimator used here[49] relies on virial factors based on AGN luminosities of $10^{42-45}$ erg s$^{-1}$, which largely overlap with our sample but are based on Doppler line widths with FWHM > 1,000 km s$^{-1}$ (for example, see ref. 76), unlike most objects here. Despite this difference, our sample is consistent with the $M_{BH}$–$\sigma_\star$ relation in ref. 77 within 1–2$\sigma$, except for objects I, J, K and L, which are below the relation by 2–3$\sigma$. This may indicate that the relation may have a greater scatter for this population or even may not be in place, yet for some systems, if they are the earliest SMBHs. Furthermore, objects in our sample probably have a BLR covering factor close to unity (that is, the fraction of the solid angle of a sphere, $\Omega/4\pi$) compared with the typically assumed values 0.1–0.5 (ref. 78), as that would help to explain the weakness of X-ray and radio emission that occurs ubiquitously[10] (see the subsequent sections and the main text). Thus, if the factor is 10 times greater, the masses here may be overestimated by up to a factor of about 3, as they scale as $M_{BH} \propto L(H\alpha)^{0.5}$. This is comparable to a scatter of 0.5 dex of typical single-epoch estimator masses[79] and is therefore not expected to significantly affect the results. Alternatively, a typical BLR size in our sample may be different from that expected for its luminosity from the radius–luminosity relation. If it was smaller, the relation $R_{BLR} \propto L(H\alpha)^{0.47}$ (ref. 49) would have a shallower slope and even then the masses would be biased to higher values only by up to 0.5 dex for a slope value of 0.2 (based on $L(H\alpha)$ in Extended Data Table 3). To conclude, we expect that our estimates are not significantly affected by the extrapolation, but that future work is required to investigate the black hole–galaxy relation in a larger sample and the validity of the current radius–luminosity relations through multi-epoch observations of these systems with exponentially extended broad line tails.

### Spectral energy distribution modelling

To estimate stellar masses of host galaxies of our objects (Fig. 4 and Extended Data Table 1), we model their spectral energy distributions (SED) from 0.9 μm to 13 μm using JWST and HST photometry from JADES[54], CEERS[53], PRIMER[80] and GO-3577 (PI E. Egami)[81] surveys, reduced with grizli[82] and available on DJA[18,19].

We model the rest-UV-to-optical SED with the code Bayesian Analysis of Galaxies for Physical Inference and Parameter EStimation (BAGPIPES)[83]. In this work, we model the sources in our sample with a double power-law star-formation history (SFH), which is characterized by distinct falling and rising slopes (see Eq. 10 in ref. 83). We impose the following priors on the SFH: $\tau \in (0, 15)$ Gyr with a uniform prior, where $\tau$ is a timescale related to the turnover between the falling and rising components of the SFH, $\alpha \in (0.01, 1,000)$ and $\beta \in (0.01, 1,000)$ with logarithmic priors, where $\alpha$ and $\beta$ are the falling and rising exponents of the SFH, respectively, $\log(M_{formed}/M_\odot) \in (6, 13)$ with a uniform prior, where $M_{formed}$ is the total mass formed and $Z_\star/Z_\odot \in (0.01, 1.2)$, where $Z_\star$ is the stellar metallicity. Moreover, we choose a 'Calzetti' dust attenuation law[84] with $A_V \in (0, 6)$ mag with a uniform prior, a nebular ionization parameter of $\log U \in (-4, -0.01)$ with a uniform prior and an intrinsic line velocity dispersion of $v \in (50, 500)$ km s$^{-1}$ with a uniform

prior. We fix the redshift of each object to the spectroscopic redshift derived using msaexp[63].

## The effect of photoelectric absorption on the X-ray luminosity

We model the effect of photoelectric absorption by a 10% solar metallicity, ionized gas column at $z = 5$ on the observed photon flux using the absori model in Xspec[85] (v.12.14.0h) with a column density of $N_H = 5 \times 10^{24}$ cm$^{-2}$. This absorber reduces the photon flux by a factor of about two for a Milky Way-absorbed power-law model with photon index $\Gamma = 2.0$ and a factor of about three for the model with photon index $\Gamma = 3.0$ and a local equivalent hydrogen column density of $N_H = 5 \times 10^{20}$ cm$^{-2}$. The exact ionization state of the gas for an absorber in the 10,000–30,000 K range is not very important in this respect—the absorption increases by about 20% for a fully neutral gas—because these temperatures are not hot enough to liberate the L- and K-shell electrons that provide most of the X-ray opacity. Photoelectric absorption from this gas, therefore, seems insufficient to provide more than half an order of magnitude flux deficit at X-ray wavelengths[9,10].

## Data availability

This paper makes use of the JWST data downloaded from the DJA[18,19] (https://dawn-cph.github.io/dja/). We are grateful for the availability of the raw data from the respective observational programmes listed in the Methods sections 'Spectroscopic sample' and 'Spectral energy distribution modelling'. Individual data products can be identified in their respective programmes or on DJA using Survey-Field-MSAID combinations in Extended Data Table 1. Source data are provided with this paper.

## Code availability

The spectra downloaded from the DJA were processed from the original telescope data with msaexp[63]. Our Monte Carlo scattering code is available upon request. We used grizli[82] to produce composite multiband colour images in Fig. 1. This study has made use of the following publicly available packages: arviz[86], astropy[87,88], Bagpipes[83], emcee[66], grizli[82], matplotlib[89], numpy[90], pandas[91], PyMC[65], scipy[92], Xspec[85] and a P Cygni module adapted from the code of Ulrich Noebauer (https://github.com/unoebauer/public-astro-tools[93,94]) and presented in refs. 95,96. The electron-scattering Monte Carlo module and relation between the line width and optical depth of scattering can be found at GitHub (https://github.com/rasmus98/ElectronMC). We make our spectroscopic line fitting routine available at GitHub (https://github.com/vadimrusakov/LRD_broad_lines).

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

**Acknowledgements** We thank G. Mazzolari for the inspiring discussions. We acknowledge support from the Danish National Research Foundation under grant no. DNRF140. V.R., T.H., N.A., D.A., C.J.C. and C.M.G. are funded by the ERC Advanced Investigator Grant EPOCHS (788113). D.W., G.P.N., R.D., S.A.S. and A.S. are co-funded by the European Union (ERC, HEAVYMETAL, 101071865). R.D. is co-funded by the Villum Foundation. K.E.H. acknowledges funding from the Swiss State Secretariat for Education, Research and Innovation under contract no. MB22.00072. S.A.S. acknowledges funding from the UK Science and Technology Facilities

Council (grant no. ST/X00094X/1). Views and opinions expressed are, however, those of the authors only and do not necessarily reflect those of the European Union or the European Research Council. Neither the European Union nor the granting authority can be held responsible for them. The data products presented in this study were retrieved from the DAWN JWST Archive (DJA). DJA is an initiative of the Cosmic Dawn Center (DAWN), which is funded by the Danish National Research Foundation under grant DNRF140.

**Author contributions** V.R. and D.W. wrote the paper and produced the figures. V.R., D.W. and G.P.N. analysed the spectroscopic data and tested the models. G.B. and K.E.H. reduced the spectroscopic and photometric data. R.G. and T.H. analysed the photometric and low-resolution spectral data and inferred the stellar masses. D.W., S.A.S., R.D. and A.S. produced the electron-scattering and P Cygni numerical models. A.P.V., V.R. and D.A. produced stellar population and photoionization models and made comparisons with the data. N.A., D.A., C.J.C., C.M.G., S.T. and J.W. helped to improve the paper and provided useful comments on the hypothesis, interpretation of the results and the wider context of supermassive black hole and galaxy evolution. All authors reviewed and edited the paper.

**Competing interests** The authors declare no competing interests.

**Additional information**
**Correspondence and requests for materials** should be addressed to V. Rusakov.

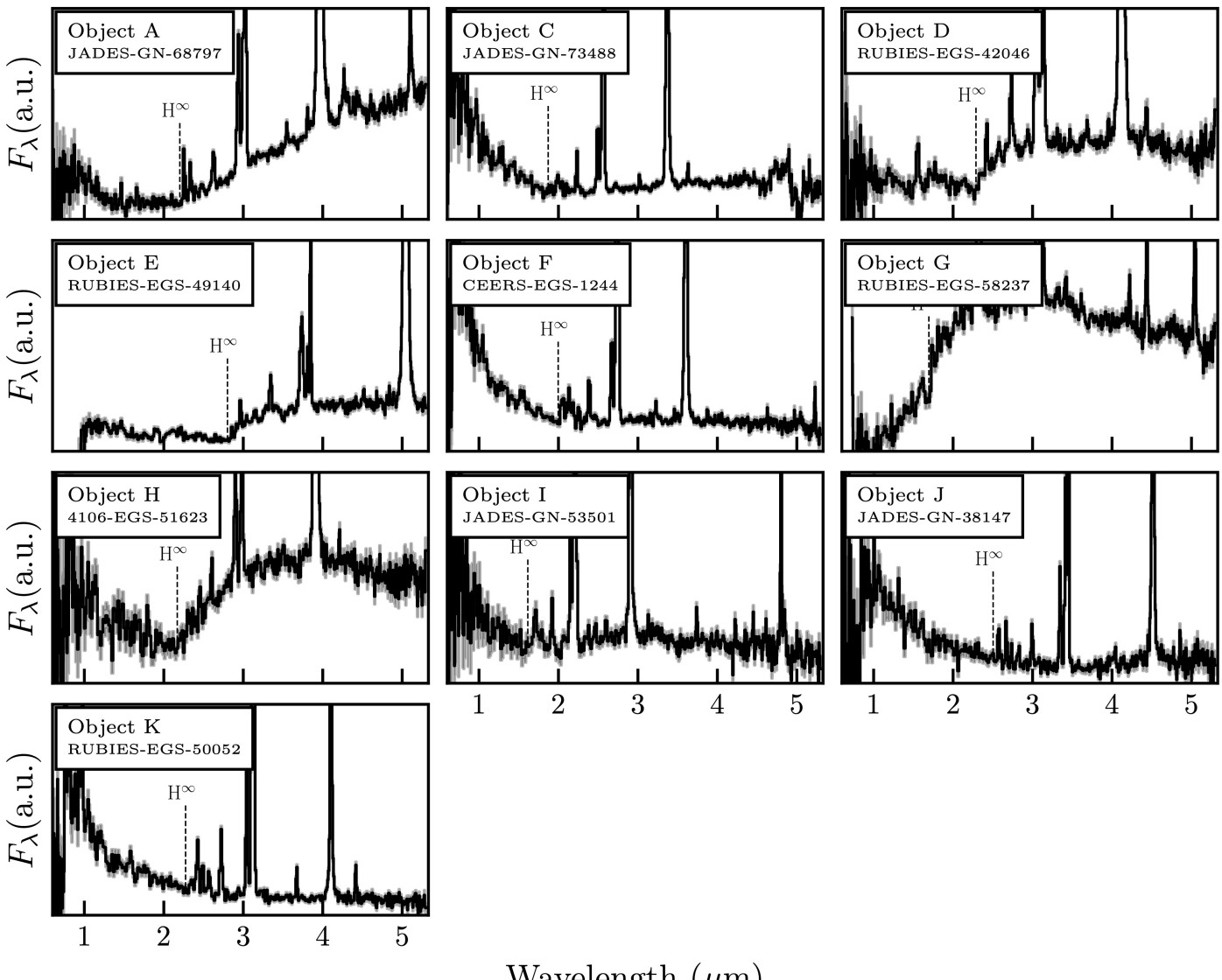

**Extended Data Fig. 1 | A gallery of NIRSpec/PRISM spectra for the high-SNR sample.** Note, objects B and L have not been observed with PRISM. The flux density is scaled arbitrarily to emphasize the continuum shapes. The position of the Balmer series limit at about 365 nm is labelled on the panels as $H^\infty$.

Interestingly, objects D, E, G, and H show clear Balmer break features, while objects A, C, F, and I have a turnover in the continuum slope coincident with the location of $H^\infty$. These SEDs are also similar to those typical of LRDs.

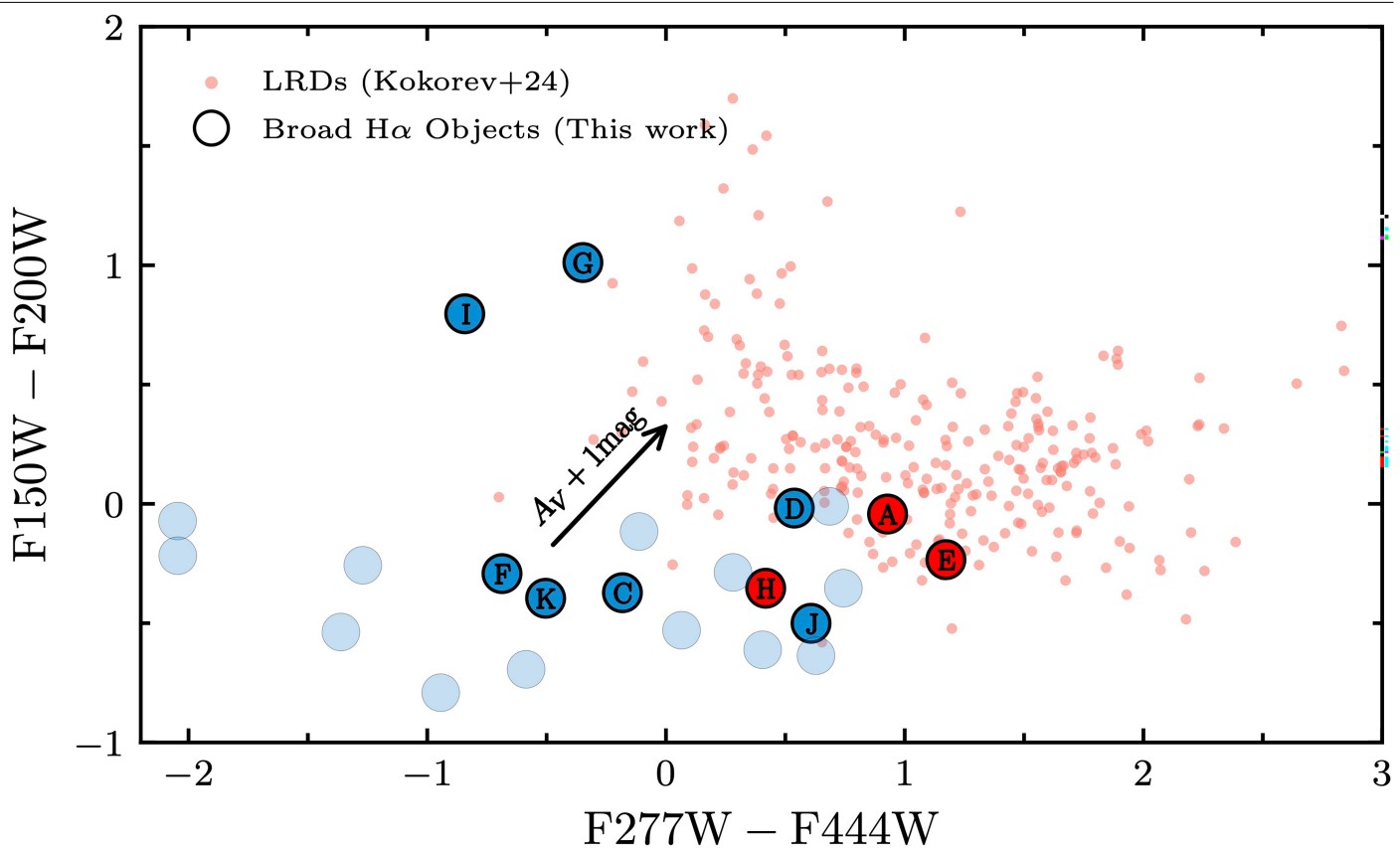

**Extended Data Fig. 2 | Colour space (rest optical versus rest UV) of the broad Hα sample in this work (large circles).** It is plotted next to the photometrically-identified population of LRDs from[97]. The objects from our sample that satisfy their colour classification criteria of LRDs (we used only the colour criteria but not the compactness criterion) are shown in red; the rest are in blue (the objects used in the stack are shown as faint blue circles). The high-SNR object B is missing a PRISM spectrum, while object L is missing due to lack of coverage in the rest UV, but has F277W − F444W = 0.24. Objects G and I are the lowest redshift objects in our sample, which may explain their redness in F150W − F200W. Some of our selection is somewhat bluer in the rest UV and optical than the population of LRDs. The difference may be explained by host galaxy contributions and dust extinction. The black arrow shows the mean effect of an extinction of $A_V$ = +1 mag on the PRISM spectra of our sample.

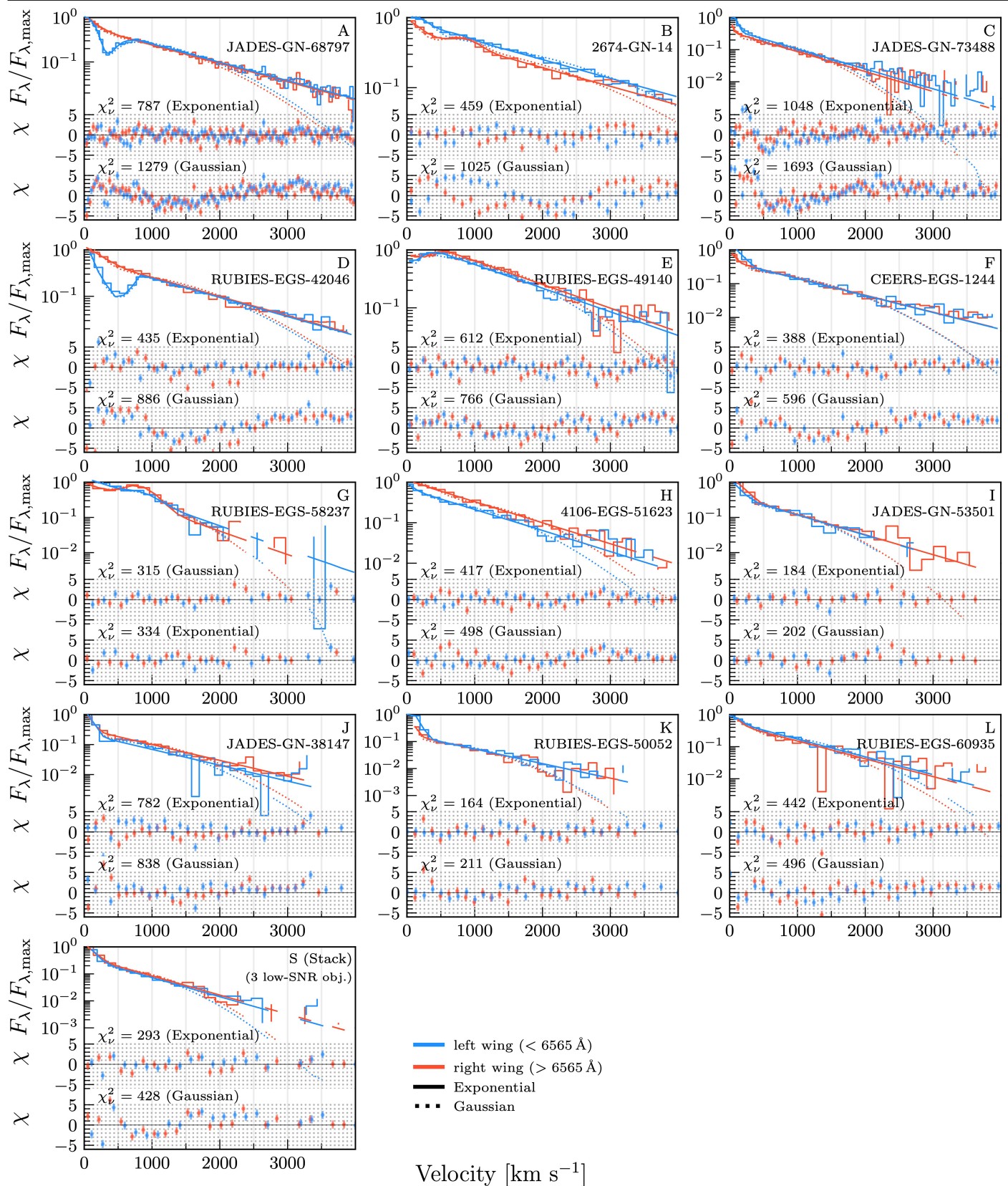

**Extended Data Fig. 3 | Comparison between the best Hα profiles modelled with a Gaussian or an Exponential broad-line component.** The wings of the Hα profiles are mirrored around the centres to demonstrate their symmetry. The objects appear in order of decreasing SNR(Hα) and their flux density is normalized for comparison. The residuals of the best-fitting models are shown below the emission line plots in terms of the number of standard deviations. In all cases, except G, and in the stack, the exponential (solid lines) is strongly preferred over the typically assumed Gaussian profile (dotted lines).

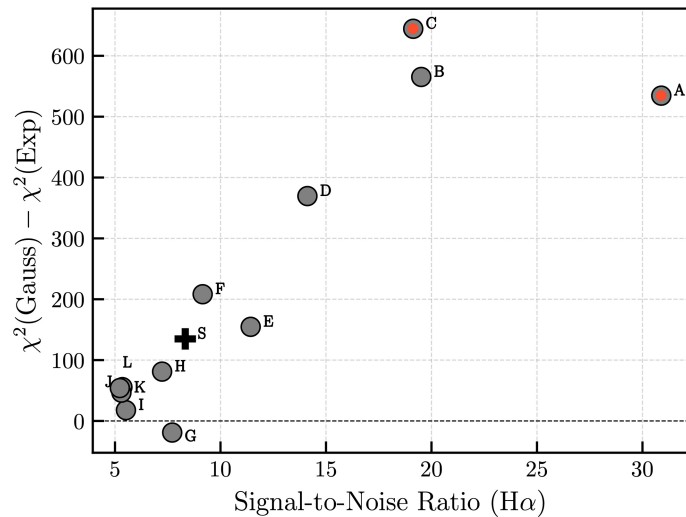

**Extended Data Fig. 4 | Comparison between the best Hα line models that include basic exponential and Gaussian broad-line components.** This shows an improvement in the goodness-of-fit of the basic exponential line shape over the Gaussian in terms of $\chi^2$ differences with increasing SNR. The plus sign indicates the stacked spectrum, and two red circles represent the high-resolution spectra.

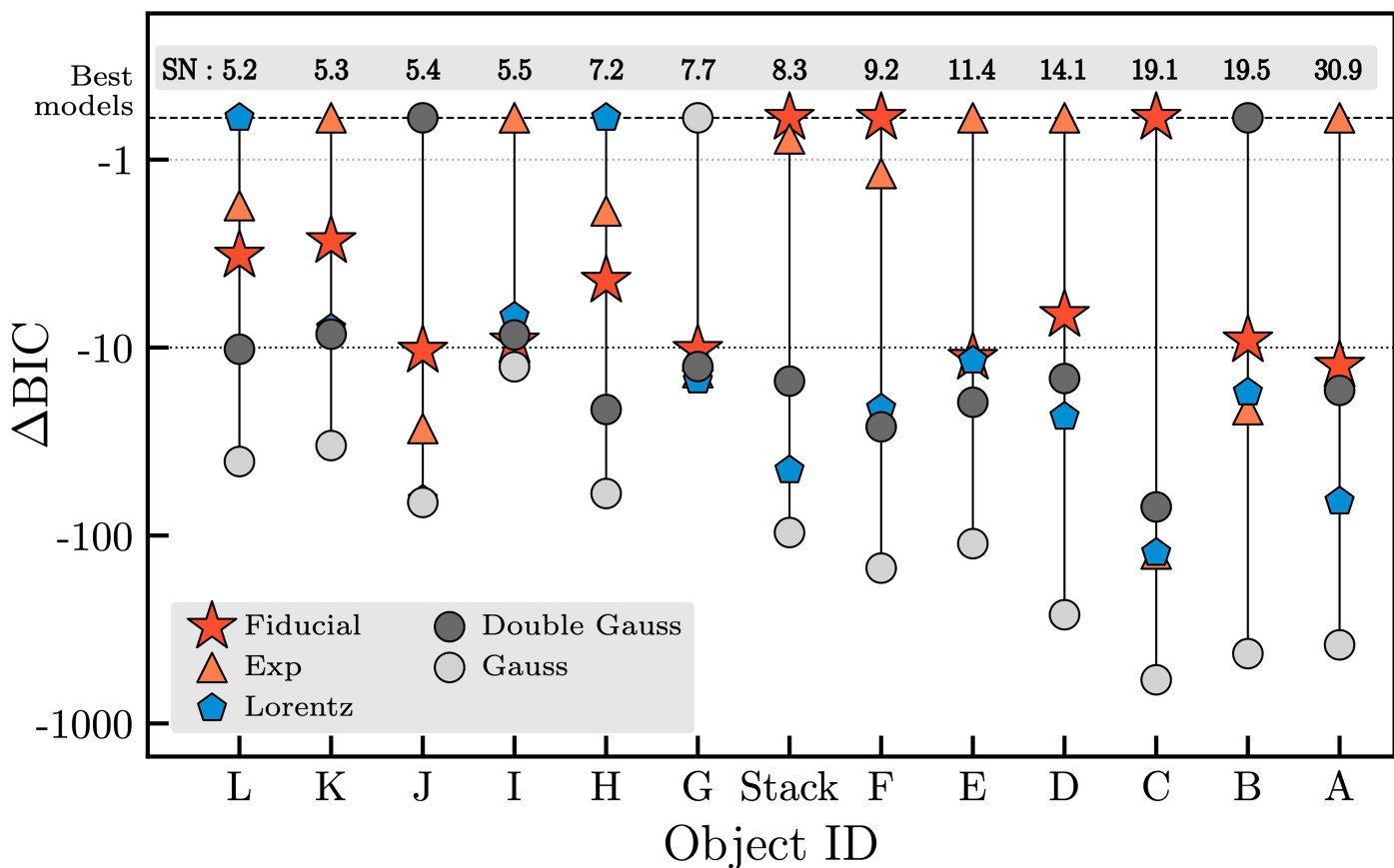

**Extended Data Fig. 5 | Differences between the Bayesian Information Criterion (BIC) values of all broad Hα models with respect to the best one (horizontal dashed line).** The broad line models include Fiducial, and Exponential, basic Lorentzian, Gaussian and Double-Gaussian. The objects are sorted from left to right in the order of increasing median SN(Hα) (the values are shown at the top). All models have equal degrees of freedom per object, except Fiducial, which has one or two additional parameters, and Double-Gaussian which has three additional parameters. This difference translates to ΔBIC=10 for the typical number of data points–therefore, for all objects the Fiducial model is either the best-fit model or statistically equivalent to the best one. Where the pure Exponential model is more favoured by ΔBIC, this indicates that the additional components of the fiducial model are not statistically required in those cases. Exceptions are object G, where the Gaussian may be marginally preferred or B and J, where the Double-Gaussian may be preferred. In objects H and L, the Lorentzian is very marginally preferred, but this is not statistically significant compared to the exponential models. In case of objects A and D, the global minimum is challenging to recover due to the P Cygni feature. We notice that the uncertainties in the spectra may be underestimated and therefore scale all chi-squared values used for calculating BIC in this figure to the single lowest reduced chi-square of $\chi_\nu^2 = 1.4$ to correct for that, assuming $\chi_\nu^2 = 1.4$ corresponds to an ideal best solution. Our best-fit model selections do not change after this correction.

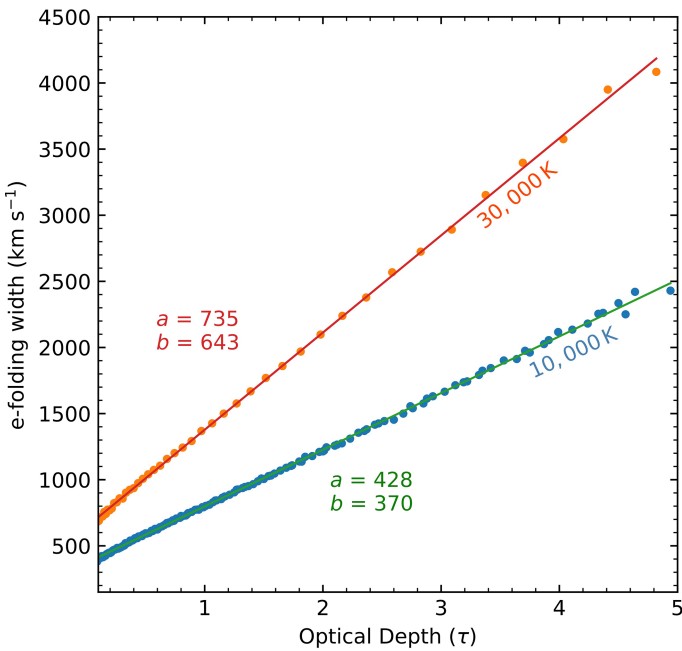

**Extended Data Fig. 6 | Relation between the optical depth of the scatterer and the resulting line broadening.** We plot here the results of Monte Carlo slab scattering simulations at electron temperatures of 10,000 K and 30,000 K.

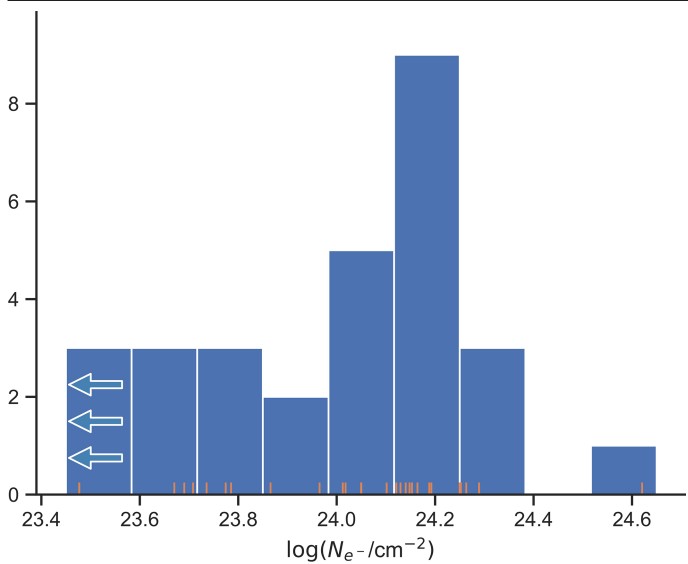

**Extended Data Fig. 7 | The distribution of free electron column densities inferred from the exponential fits to the Hα line for our full sample.** The first column of the histogram represents upper limits. Our sample selection of FWHM ≥ 1000 km s$^{-1}$ limits the lowest column densities we can detect. Objects with column densities log($N_{e^-}$/cm$^2$) ≥ 24.5 are less likely to be discovered as they correspond to optical depths $\tau_e$ > 2. The intrinsic distribution of the column densities of all sources is therefore likely to be broader than shown here.

**Extended Data Table 1 | Derived properties of the broad Hα systems studied in this work based on the best-fit fiducial model**

| ID | Survey-Field-MSAID | $\alpha$ (°) | $\delta$ (°) | Redshift | $\tau$ | $N_{e^-}$ $(10^{24}\,\mathrm{cm}^{-2})$ | $M_{BH,F}$ | $M_{BH,G}$ | $M_\star$ |
|---|---|---|---|---|---|---|---|---|---|
| | | | | | | | | | —— (log $M_\odot$) —— |
| A | JADES-GN-68797 | 189.2291375 | 62.1461897 | 5.0405 | $1.2^{+0.2}_{-0.1}$ | $1.8^{+0.3}_{-0.1}$ | $6.4^{+1.0}_{-0.2}$ | 8.8 | $< 10.9$ |
| B | 2674-GN-14 | 189.1998125 | 62.1614742 | 5.1826 | $2.8^{+0.3}_{-0.3}$ | $4.2^{+0.5}_{-0.4}$ | $7.9^{+0.1}_{-0.1}$ | 8.5 | $< 10.2$ |
| C | JADES-GN-73488 | 189.1973958 | 62.1772331 | 4.1327 | $0.9^{+0.1}_{-0.1}$ | $1.3^{+0.2}_{-0.1}$ | $6.1^{+0.1}_{-0.1}$ | 7.5 | $< 9.6$ |
| D | RUBIES-EGS-42046 | 214.7953667 | 52.7888467 | 5.2757 | $1.5^{+0.3}_{-0.2}$ | $2.3^{+0.4}_{-0.2}$ | $7.6^{+0.1}_{-0.1}$ | 9.0 | $< 10.3$ |
| E | RUBIES-EGS-49140 | 214.8922458 | 52.8774097 | 6.6847 | $1.3^{+0.2}_{-0.1}$ | $1.9^{+0.2}_{-0.2}$ | $< 6.4$ | 8.5 | $< 10.6$ |
| F | CEERS-EGS-1244 | 215.2406542 | 53.0360411 | 4.4771 | $1.0^{+0.1}_{-0.1}$ | $1.5^{+0.2}_{-0.1}$ | $5.9^{+0.3}_{-0.2}$ | 7.9 | — |
| G | RUBIES-EGS-58237 | 214.8505708 | 52.8660297 | 3.6505 | $< 0.8$ | $< 1.2$ | $7.5^{+0.1}_{-0.1}$ | 7.5 | $< 10.6$ |
| H | 4106-EGS-51623 | 214.8868167 | 52.8553892 | 4.9511 | $1.0^{+0.1}_{-0.1}$ | $1.5^{+0.2}_{-0.2}$ | $6.6^{+0.1}_{-0.2}$ | 7.8 | $< 10.1$ |
| I | JADES-GN-53501 | 189.2950583 | 62.1935722 | 3.4294 | $0.7^{+0.2}_{-0.1}$ | $1.1^{+0.2}_{-0.2}$ | $< 5.6$ | 7.4 | $< 9.6$ |
| J | JADES-GN-38147 | 189.2706750 | 62.1484186 | 5.8694 | $0.9^{+0.1}_{-0.1}$ | $1.4^{+0.2}_{-0.2}$ | $5.6^{+0.2}_{-0.3}$ | 7.8 | $< 8.9$ |
| K | RUBIES-EGS-50052 | 214.8234542 | 52.8302767 | 5.2392 | $0.9^{+0.1}_{-0.1}$ | $1.3^{+0.2}_{-0.2}$ | $5.9^{+0.2}_{-0.2}$ | 7.5 | $< 9.5$ |
| L* | RUBIES-EGS-60935 | 214.9233750 | 52.9255928 | 5.2877 | $0.9^{+0.2}_{-0.2}$ | $1.4^{+0.3}_{-0.2}$ | $< 5.4$ $6.7^{+0.3}_{-0.2}$ | 7.8 | $< 9.4$ |
| S† | — | — | — | — | $0.5^{+0.1}_{-0.1}$ | $0.7^{+0.1}_{-0.1}$ | — | — | $< 8.9$ |

We report the estimates of the optical depth, $\tau$, and associated gas column densities, $N_{e^-}$, based on our electron-scattering gas model and the exponential line widths. We estimate SMBH masses using the relation of ref. 34 based on intrinsic narrow line cores in our fiducial model ($M_{BH,F}$) or based on single Gaussian BLR model ($M_{BH,G}$). Stellar mass estimates from best-fit photometric SED models, are denoted as upper limits because the likely dominant contribution from the AGN's nebular emission to the restframe optical continuum is neglected. Objects ABFGIJ are reported in this work, as extracted from DJA. References to other sources where the objects have been previously reported are as follows: C[4], E[98], DEHL[16], K[3]. The stack contains spectra from most of these studies. The typical uncertainties in the redshifts are about $3 \times 10^{-4}$. Right ascensions and declinations are relative to J2000.0. *Object L has two estimates of $M_{BH}$ corresponding to two distinct FWHM solutions. †$M_\star$ for the stack is a median of the individual masses.

**Extended Data Table 2 | Goodness of fit statistics of all models**

| ID | Hα SNR (/10 Å) | n | Other | k Fiduc. | 2-Gauss | BIC Gauss | 2-Gauss | Lorentz | Expo. | Fiduc. | $\chi^2$ Gauss | 2-Gauss | Lorentz | Expo. | Fiduc. |
|---|---|---|---|---|---|---|---|---|---|---|---|---|---|---|---|
| A* | 3.1 | 491 | 12 | 14 | 15 | 991.6 | 626.7 | 675.7 | 609.8 | 622.3 | 917.3 | 533.7 | 601.3 | 535.4 | 535.5 |
| B | 2 | 276 | 9 | 11 | 12 | 782.5 | 357.4 | 374.7 | 378.8 | 366.5 | 731.9 | 289.9 | 324.1 | 328.2 | 304.7 |
| C* | 1.9 | 483 | 8 | 10 | 11 | 1258.5 | 743.2 | 796.4 | 798.2 | 672.7 | 1209.1 | 675.2 | 746.9 | 748.8 | 610.9 |
| D | 1.4 | 198 | 12 | 14 | 15 | 580.7 | 331.3 | 340.1 | 316.7 | 323.4 | 517.2 | 252.0 | 276.6 | 253.3 | 249.4 |
| E | 1.1 | 313 | 11 | 13 | 14 | 610.6 | 519.7 | 511.8 | 500.1 | 511.5 | 547.4 | 439.2 | 448.6 | 436.9 | 436.8 |
| F | 0.9 | 224 | 8 | 10 | 11 | 468.9 | 346.2 | 340.9 | 320.2 | 319.8 | 425.6 | 286.7 | 297.6 | 276.9 | 265.7 |
| G | 0.8 | 155 | 9 | 11 | 12 | 270.1 | 282.7 | 285.1 | 283.7 | 280.4 | 224.7 | 222.2 | 239.7 | 238.3 | 224.9 |
| H | 0.7 | 280 | 8 | 10 | 11 | 401.0 | 362.5 | 341.2 | 343.0 | 345.6 | 355.9 | 300.5 | 296.1 | 298.0 | 289.2 |
| I | 0.6 | 103 | 8 | 10 | 11 | 181.1 | 177.0 | 175.3 | 168.4 | 177.7 | 144.0 | 126.0 | 138.2 | 131.3 | 131.4 |
| J | 0.5 | 403 | 8 | 10 | 11 | 646.3 | 579.6 | 644.1 | 606.6 | 590.0 | 598.3 | 513.6 | 596.1 | 558.6 | 530.0 |
| K | 0.5 | 120 | 8 | 10 | 11 | 188.7 | 164.0 | 163.4 | 155.5 | 158.2 | 150.4 | 111.3 | 125.1 | 117.2 | 110.3 |
| L | 0.5 | 222 | 8 | 10 | 11 | 397.6 | 367.4 | 357.2 | 359.0 | 360.5 | 354.4 | 308.0 | 314.0 | 315.8 | 306.4 |
| S | 0.8 | 154 | 7 | 9 | 10 | 341.1 | 259.8 | 289.7 | 244.8 | 244.7 | 305.9 | 209.4 | 254.4 | 209.6 | 199.4 |

The statistics include the Bayesian Information Criterion (BIC) and $\chi^2$ values. Each model uses $n$ data points and $k$ free parameters. *Analysis of objects A and C is based on spectra taken with the high-resolution G395H grating, while all others use the medium-resolution G395M. SNR (Hα) is the median SNR within 2000 km s$^{-1}$ of the Hα λ6564.6 wavelength. We notice that the uncertainties in the spectra may be underestimated and therefore scale all chi-squared values in this table (BIC values are modified as a result) to the single lowest reduced chi-square of $\chi_\nu^2 = 1.4$ to correct for that, assuming $\chi_\nu^2 = 1.4$ corresponds to an ideal best solution.

**Extended Data Table 3 | Properties of the broad component of Hα based on the best-fit fiducial model**

| | Broad H$\alpha$ Component | | | | | |
|---|---|---|---|---|---|---|
| ID | EW [Å] | $F \times 10^{-19}$ [erg s$^{-1}$ cm$^{-2}$] | $\log L(\mathrm{H}\alpha)$ | FWHM$_{\mathrm{Exp}}$ [km s$^{-1}$] | FWHM$_{\mathrm{Doppler}}$ [km s$^{-1}$] | FWHM$_{\mathrm{instrum}}$ [km s$^{-1}$] |
| A | $1106^{+49}_{-60}$ | $1814^{+74}_{-23}$ | $43.70^{+0.02}_{-0.01}$ | $1451^{+26}_{-28}$ | $315^{+667}_{-73}$ | 66 |
| B | $410^{+6}_{-6}$ | $774^{+9}_{-10}$ | $43.36^{+0.01}_{-0.01}$ | $2551^{+109}_{-101}$ | $2052^{+128}_{-148}$ | 171 |
| C | $1124^{+42}_{-40}$ | $273^{+3}_{-3}$ | $42.68^{+0.00}_{-0.00}$ | $1162^{+29}_{-26}$ | $398^{+24}_{-25}$ | 77 |
| D | $1654^{+60}_{-66}$ | $1344^{+16}_{-22}$ | $43.62^{+0.01}_{-0.01}$ | $1684^{+113}_{-84}$ | $1494^{+125}_{-138}$ | 168 |
| E | $1533^{+103}_{-141}$ | $2094^{+245}_{-30}$ | $44.05^{+0.05}_{-0.01}$ | $1474^{+34}_{-33}$ | $< 274$ | 137 |
| F | $1238^{+51}_{-45}$ | $583^{+12}_{-15}$ | $43.09^{+0.01}_{-0.01}$ | $1235^{+28}_{-27}$ | $248^{+82}_{-115}$ | 193 |
| G | $201^{+18}_{-21}$ | $340^{+27}_{-34}$ | $42.64^{+0.03}_{-0.05}$ | $< 1003$ | $1957^{+82}_{-78}$ | 227 |
| H | $733^{+25}_{-24}$ | $240^{+4}_{-5}$ | $42.80^{+0.01}_{-0.01}$ | $1266^{+64}_{-61}$ | $643^{+107}_{-103}$ | 177 |
| I | $1038^{+335}_{-210}$ | $228^{+58}_{-10}$ | $42.41^{+0.10}_{-0.02}$ | $1061^{+85}_{-69}$ | $< 274$ | 238 |
| J | $1106^{+68}_{-60}$ | $265^{+8}_{-8}$ | $43.02^{+0.01}_{-0.01}$ | $1215^{+40}_{-44}$ | $197^{+42}_{-56}$ | 154 |
| K | $996^{+186}_{-141}$ | $186^{+10}_{-11}$ | $42.75^{+0.02}_{-0.03}$ | $1171^{+60}_{-66}$ | $311^{+80}_{-70}$ | 169 |
| L | $1257^{+123}_{-112}$ | $361^{+13}_{-14}$ | $43.05^{+0.02}_{-0.02}$ | $1208^{+84}_{-117}$ | $< 155$ $627^{+203}_{-122}$ | 168 |
| S$^{(\mathrm{Stack})}$ | — | — | — | $871^{+28}_{-24}$ | $339^{+55}_{-51}$ | 246 |

In the fiducial model the broad Hα component is represented as a combination of an intrinsic Gaussian convolved with an exponential kernel and a pure Gaussian (electron-scattered and unscattered components), where the best-fit widths of the intrinsic Gaussian and convolved exponential components are reported as FWHM$_{\mathrm{Doppler}}$ and FWHM$_{\mathrm{Exp}}$. The Gaussian line widths (FWHM$_{\mathrm{Doppler}}$) in this table are corrected for instrumental broadening (FWHM$_{\mathrm{instrum}}$). In the case of object L, the table shows two distinct posterior solutions for FWHM$_{\mathrm{Doppler}}$.