## [Peer Review File · Nature]

Little red dots as young supermassive black holes in dense ionized cocoons

Corresponding Author: Dr Vadim Rusakov

Version 0:

Reviewer comments:

Referee #1

(Remarks to the Author)

Report on the paper "JWST's little red dots: an emerging population of young, low-mass AGN cocooned in dense ionized gas", by Rusakov et al.

The authors present an interesting analysis of "little red dots" (LRDs) - a class of high-redshift objects discovered by JWST that potentially harbor supermassive black holes (SMBHs). Using JWST NIRSpec observations of 13 sources, they argue that emission line profiles favor exponential shapes over Gaussian profiles, suggesting that electron scattering, rather than gas motion, is the primary mechanism for line broadening. They argue that this interpretation leads to considerably lower black hole mass estimates than previously reported. The authors propose that LRDs consist of low-mass SMBHs surrounded by obscuring Compton-thick gas that suppresses radio and X-ray emission, accreting at near-Eddington rates—effectively making them narrow-line Seyfert 1 analogs at high redshift. They argue that this model would help explain several puzzling aspects of recent LRD observations, including SMBHs that appear too massive for their cosmic age and their lack of expected multiwavelength counterparts.

While the authors present an interesting alternative scenario for interpreting LRDs, the physical interpretation requires further justification as well as an improvement in the statistical evidence analysis before the conclusions can be considered robust. The authors should more thoroughly consider alternative explanations based on established AGN studies and provide clearer statistical evidence for their preferred model, answering to the aspects/concerns listed below.

1. Statistical Evidence for Exponential Profiles

The establishment of exponential line profiles over Gaussian profiles is crucial for the paper's subsequent interpretations. While I appreciate the detailed comparison in the Appendix (Figure 7 and Table 3), the main text lacks a clear statistical summary of model preference across the entire sample. The BIC differences cited ($|\Delta\text{BIC}| > 10$) for high S/N objects do suggest evidence favoring the exponential or fiducial model, but a more comprehensive statistical statement is needed.

The authors should:

- Provide a clear summary of which model is preferred across the whole sample (e.g., "Model B was identified as the best model in X out of Y spectra")
- Transform ΔBIC values into model probabilities or BIC weights for increased interpretability
- Provide average weights across the dataset to quantify the overall statistical preference

2. Alternative Explanations Based on Known AGN Properties

The paper's interpretation conflicts with established knowledge of AGN emission line profiles in the nearby universe. Local AGN typically show: (i) A narrow component from low-density clouds (the Narrow-Line Region or NLR) with widths of a few hundred km/s extending to distances of a few hundred pc from the nucleus; (ii) A broad component from high-density clouds (the Broad-Line Region or BLR) with widths above ~ 2000 km/s at sub-pc distances.

Recent studies have shown that: (i) NLR profiles often require multiple Gaussian components, sometimes including a

narrow and a broader component associated with outflows; (ii) BLR emission-line profiles are highly variable and rarely well-reproduced by a single Gaussian, often reflecting particular geometries (e.g., flattened rings) and kinematics involving outflows or inflows, a scenario confirmed by reverberation mapping studies.

Thus, although there may be the presence of electron scattering in the BLR, as argued by the authors, their test does not consider the possibility that the BLR may present a different profile than a Gaussian. They only compare the residuals of the fit of their model with that of a Gaussian.

3. Interpretation of Line Components

A close examination of Figure 3 reveals patterns that may better align with conventional AGN interpretations:

- a) Their fits of the H α profiles of targets B and G are the only ones that present broad-line profiles (black line fits in Fig. 3) that have similar widths to those of the BLR of type 1 AGN in the near-Universe; the other 11 sources are argued to have very narrow BLR profiles (as shown by the black line fits);
- b) The authors argue that the profiles of the NLR are represented by only one narrow Gaussian (in blue in Fig. 3); but when I look at their fits, what they call the BLR component in the 11 sources (excluding B and G) has a similar or slightly larger width than that of the NLR, and could be identified with the second component of the NLR (as in the near-Universe), instead of arguing that it would come from the BLR;
- c) Inspection of the emission-line profiles of sources C, F, H, I, J and L show changes of curvature in the profile between the narrow core (from the NLR) and broad wings (from the BLR) that suggest that the wings are part of another component, as in the near-Universe: another set of clouds from the BLR that are denser, closer to the SMBH and move faster than those of the NLR.
- d) In line 49, the authors argue: "... relatively narrow Balmer absorption features are also a common feature in these objects"... But Fig. 3 shows this absorption in only on 3 of the 13 objects discussed, thus does not seem so common!
- e) In line 83, it is argued: "... The absence of strong broadening of the [O III] $\lambda\lambda$ 4959, 5007 doublet, while the H β line next to it is broadened, suggests that the electron volume density, n_e , in the scattering region, must be at least several times the critical density for these lines."... This is also the case in the usual interpretation of the BLR - origin of the broad H β component, due to its higher density, supporting that the BLR and NLR in the LRDs present similar properties to those of nearby AGN.
- f) The authors argue that the widths of the lines imply low black hole masses, on the assumption that the virial velocities is given by the widths of the lines. It is not clear if the authors are arguing that one should use the BLR width that they propose (the ones corresponding to the black profiles in Fig. 3 and listed under the profiles). These are too narrow; as pointed out above, most of them have widths typical of the NLR, while the virial relations use the actually measured FWHM of the broad profiles, not assuming a narrow profile and broadening it via electron scattering.
- g) Remark regarding Fig. 3: The labeling is partly difficult to read and interpret. For example, the meaning of "Gauss*Exp" is not immediately clear, neither that the black profile is that of the BLR component supposed to be scattered. I understood this only after reading the whole paper. Improved labeling and explanation (or perhaps separating some elements into additional figures) would enhance clarity.

4. Physical Plausibility

The authors propose that Compton-thick ionized gas surrounding the SMBH explains the lack of radio and X-ray emission in LRDs. However:

- The likelihood of Compton-thick obscuring structure being fully ionized at scales of ~ 100 light days requires further justification.
- Even with such obscuration, one might expect parsec-scale jets whose emission would not be obscured by the proposed material.
- The possibility that these objects are intrinsically radio-quiet should be more thoroughly considered.

Referee #2

(Remarks to the Author)

I co-reviewed this manuscript with one of the reviewers who provided the listed reports.

Referee #3

(Remarks to the Author)

This paper presents a spectroscopic analysis of 12 broad-line galaxies (with H α FWHM > 1000 km/s), using high signal-to-noise (SNR > 5 near H α) JWST/NIRSpec medium-resolution spectra. The dataset includes observations from CEERS, JADES, RUBIES, and 18 stacked spectra, spanning a redshift range of 3.4–6.7. For two of these galaxies, high-resolution spectra were also available. The authors argue that the observed line profiles are better fit by an exponential profile—consistent with Compton scattering in a dense medium—combined with a narrow Gaussian, rather than a single Gaussian profile typically associated with virialized gas motion around a supermassive black hole (SMBH). If correct, this result implies that previously reported black hole (BH) masses may be significantly overestimated and accretion rates underestimated, which would have substantial implications for models of SMBH formation and the interpretation of JWST-detected broad-line AGNs.

The paper proposes an intriguing and potentially important reinterpretation of broad-line profiles in JWST AGNs, raising fundamental questions about black hole mass estimates in the early universe. However, the manuscript requires major

revisions before it can be considered for publication. The following concerns should be addressed:

Major Comments:

Alternative Line Profile Fits:

The authors dismiss double-Gaussian fits on the basis that such profiles would indicate dual AGNs, which they argue is unlikely for the full sample. However, broad wings in AGN line profiles have been previously reported (e.g., Nagao et al. 2006; Santos et al. 2025) and successfully modeled using double-Gaussian components. These features are often interpreted as arising from complex BLR dynamics, such as a combination of rotation and turbulence or distinct inner and outer BLR zones. The authors should revisit these alternative fits, incorporate a discussion of prior work, and evaluate how such interpretations might also lead to systematic overestimates in BH mass across AGN populations. Furthermore, they should justify why the exponential profile is favored in their data, but not in previous studies. A clear explanation of how the JWST broad-line AGNs differ—e.g., in terms of X-ray and radio properties—would strengthen the case for a distinct physical scenario.

High-Ionization Lines:

The manuscript would benefit from a more thorough analysis of the rest of the spectrum, particularly with respect to high-ionization lines. The authors should comment on whether such lines are present or absent in their sample and, if absent, provide upper limits. A comparison with previously characterized JWST AGNs—some of which show such lines and others that do not—would add valuable context.

Redshift-Dependent Cocoon Hypothesis:

The authors propose that dense, ionized gas cocoons are responsible for the exponential wings, but such structures are not observed in a significant population of lower-redshift AGNs (e.g., NLS1s). The paper should explore possible reasons for this redshift dependence. For instance, could metal-poor environments at high redshift play a role? Why are no local analogs found in similarly low-metallicity galaxies today? A more speculative discussion, while clearly labeled as such, would be appropriate here.

Comparison with Local NLS1s and High Accretion AGNs:

A more comprehensive comparison with narrow-line Seyfert 1 galaxies and other examples of high-Eddington accretors in the local universe would strengthen the paper. The authors should consider outflows seen in NLS1s and discuss why similar outflows in these high-redshift highly accreting systems may not be seen.

Missing Intermediate Population:

The authors argue for a distinct population of JWST broad-line AGNs, but the lack of a transitional or intermediate population (with lower column densities) bridging the known high-redshift quasars and these new sources raises questions. Some discussion is needed on why such a population has not been detected and what this implies about AGN evolution, duty cycles, or observational biases.

Minor Comments:

Figure 2 should include a physical scale in kiloparsecs (kpc) for reference.

While case B recombination does not strictly apply at the gas densities inferred, it would be useful to provide the observed range of recombination line ratios and compare them to those seen in typical quasar populations.

Referee #4

(Remarks to the Author)

The authors present a compelling list of arguments to interpret JWST's compact, distant galaxies showing broad lines. The H-alpha line broadening from scattered light instead of random motion in the broad line region is convincing, and so are the implications of Compton-thick ionized gas.

My main remaining concern is that the sample comes from a different color selection than typical LRDs (e.g., Extended Data Figure 2). The current work tries to explain LRDs with bluer galaxies than the LRDs. Could the authors check or comment on whether the scattered light interpretation for the broad H-alpha emitters could fully apply to the general populations of LRDs that appear redder on the color-color diagram? I would like to see at least a discussion on the potential impact (or the absence of it) of the sample selection. I assume if the majority of LRDs are missed by this work due to higher dust extinction, the line profiles could be intrinsically different. Do the spectra for A, E, H (also classified as LRDs) appear representative of the entire sample in this work?

Below are minor comments and questions.

Lines 119-120: Please correct me if I am wrong, trying to interpret the lines as a general reader: an outflow from star-formation requires 10^4 star clusters to produce 10^{45} erg/s of ionizing luminosity, but the gas masses (where do they come from, and how heavy are they?) require $>10^9$ solar luminosity per star cluster, requiring $>10^{13}$ solar luminosity for individual galaxies, ruling out the star-formation origin for the broad lines.

Lines 136-137: I thought the authors intended "consistent with the local BH-host galaxy scaling relation at lower redshifts".

Line 159 and lines 191-193: Is the spherical gas distribution a distinct feature at high- z , or in other words, are young AGN accretion seen only at high redshifts? Could the authors predict or identify any similar objects in the local universe, according to the formation mechanisms for LRDs the authors suggest?

Line 165: Some LRDs are reported with high A_V values derived from the H-alpha to H-beta ratios (Balmer decrement). If the Balmer decrement of LRDs don't follow the case B values, how much could we trust the A_V values from the Balmer decrements?

Extended Data Figure 6: I was curious if the trend became flat if the y-axis was replaced by the difference in chi-squares divided by the chi-squares of the exponential model, or replaced by the ratio of the chi-square values. If so, it would imply the exponential model fits better than the H-alpha profile at any S/N. I recall the F-test where the difference in the chi-squares divided by the reduced chi-squares of the exponential model should follow an F-distribution.

Extended Data Figure 7: Why are in some cases, the fiducial model has a poorer fit than the exponential? I guess the added degrees of freedom for the fiducial model requires a significantly improved goodness-of-fit for a larger Delta BIC value. Also, does the scaling of the reduced chi-square values to 1.4 change the Delta BIC values?

Version 1:

Reviewer comments:

Referee #1

(Remarks to the Author)

The authors answered most of the questions that we have raised. We appreciated, in particular the inclusion of the model using two Gaussian components for the BLR that they show do improve the fits, showing similar fitting goodness as their models in 5/13 of the sources. But in the others their model gave better fits. The fact that the wings show preferably exponential profiles is also in favor of their models.

We conclude that their proposed model applies to many emission-line profiles of the LRDs, but not necessarily for all the sources. In particular:

- (i) Sources B, D and G (3/12, thus 25%) show H α broad (without electron broadening) that are broader than 1500 km/s, which support the presence of a SMBH with mass $\sim 10^8$ solar masses; thus, in these cases, the width of the lines are dominated by the SMBH gravitational potential;
- (iii) The number of sources with absorption is small (3/12, the same as the number of cases cited above in (i) that support the presence of a massive SMBH);
- (ii) Sources I, J, K, L (4/12) and the Stack S in Fig. 3 seem to show H α broad component narrower than the narrow component due to the host galaxy. Does this make sense?
- (iv) Fig. 3 shows that the broad profile or several sources are somewhat asymmetric relative to the center wavelength (H α _{narrow}): we see some asymmetries (in decreasing order) in H, L, I, F, C and J.

In summary, we do agree that the presence of electron scattering is important to describe the wings of the line profiles and that their model is a good alternative to explain many LRDs, but not necessarily all.

Referee #2

(Remarks to the Author)

I co-reviewed this manuscript with one of the reviewers who provided the listed reports.

Referee #3

(Remarks to the Author)

I thank the authors for carefully addressing my comments. My comments have been adequately addressed to warrant publications with only one minor point. In the new paragraph (line 218) addressing my question about local objects, the authors should also mention the growing population of metal poor compact dwarfs that show broad lines and [NeV] emission despite being metal poor. Recent JWST observations are uncovering more of such objects where the AGN origin of the broad lines is being questioned. Some comment on this population with references to the papers (e.g. Izotov et al. 2021, Reefe et al. 2023, Hatano et al, Mingozi et al. 2025) should be mentioned.

Referee #4

(Remarks to the Author)

I now better see (correct me if I am wrong) that the fiducial model profile to fit the H-alpha lines consists of a non-scattered and a scattered (the same non-scattered Gaussian but convolved with an exponential kernel) component of an intrinsically Doppler-broadened Gaussian, and that the authors use the intrinsic Gaussian width to estimate the BH mass. Although the statistical reasoning to adopt such scattering over pure Doppler broadening appears sound, the implications of the Doppler-broadened line widths is worrisome. From the inset of Fig 3 and Table 4, the intrinsic line widths appear mostly narrower than 600 km/s, and comparable to the broadening at the narrow-line region. Moreover, objects F, I, J, K, L, S have the intrinsic Gaussian as narrow as their narrow lines, further supporting that the H-alpha line may not be originating from the broad-line region (hereafter BLR). AGN virial BH mass estimators rely on the Doppler broadening of the BLR, defined to be about 1000 km/s or broader. Could the Doppler-broadened intrinsic H-alpha profiles from the targets, be interpreted as coming from the BLR? I would rather believe that most of the targets don't have a BLR, which is a radical idea scarcely seen in existing literature on AGN evolution.

I tried to search from the literature if there are existing discussions on intermediate-line (FWHM<1000 km/s but broader than narrow lines) regions that might be responsible for explaining million-solar-mass BHs the authors try to in Fig 4 (e.g., Hu et al. 2008, ApJL, 683, 115), but couldn't find emission lines much narrower than 1000 km/s used or interpreted as broad or even intermediate. Bontà et al. (2020, ApJ, 903, 112) use reverberation mapped H-beta calibrations down to 800 km/s but not any narrower (and likewise for the low-L studies by Bentz et al. 2009, ApJ, 705, 199, and later work by their group). The apparent absence of a typically broad-line is predicted (Elitzur & Ho 2009, ApJL, 701, 91) at bolometric luminosities of around 10^{39} erg/s and below, which is much lower than the 10^{43} - 10^{45} erg/s found in this work (assuming the H-alpha is a few percent of the bolometric, line 198). All of my inquiries lead to failure in explaining the narrow line profiles of the targets.

I thus request the authors to consider the implications of the potential absence of a BLR: perhaps i) the AGN is in its earliest evolutionary phase of forming a BLR (but then I don't think the BH mass estimators could be used as the current way: perhaps an upper limit on the line widths of 1000 km/s should be imposed instead for targets with FWHM<1000 km/s), or ii) we are looking at very strange/rare AGN without the BLR, iii) the LRDs are not AGNs and the lines more likely come from star-formation, or iv) the scattering interpretation has a missing piece of the puzzle requiring further investigation.

Version 2:

Reviewer comments:

Referee #1

(Remarks to the Author)

The paper has improved, and their model provides good fits to the profiles of most of the data. But I still see a few inconsistencies and suggest small modifications to take them into account, as follows.

- In the Abstract, the authors write: "Using the highest-quality JWST spectra, we show here that the lines are broadened by electron scattering with a narrow intrinsic line core."

- To be in agreement with their findings they should add: "in 9/12 (or in most) of the studied sources", because in three sources the intrinsic line core is not narrow, but a broad component is necessary, indicating more massive black holes. Thus, at least in 25% percent of the sources, there is a massive enough black hole to broaden the line.

- Caption of Fig. 3: ..." in several cases, A P Cygni profile or a Gaussian Absorption feature are included".

- I see this only in 3/12 of the profiles, a minority, thus "several" is misleading; they should state "in 25%" (or "in some") of the sources".

- Regarding the physical scenario: The authors argue for the presence of an accreting black hole that has emission from the accretion disk cocooned in dense ionized gas. Still, there is escaping radiation to generate the NLR emission. E.g., in line 625, they write:

"Furthermore, objects in our sample likely have a BLR covering factor close to unity (i.e. the fraction of the solid angle of a sphere, $\Omega/4\pi$) compared to the typically assumed values 0.1 – 0.5, as that would help to explain the weakness of X-ray and radio emission that occurs ubiquitously (see Methods below and the main text). Thus, if the factor is ten times greater, the masses here may be overestimated by up to a factor of ~ 3 , as they scale as $M_{\text{BH}} \propto L(\text{H}\alpha)^{0.5}$ "

- 1st point I would like to make: they are calling the cocoon the BLR? Maybe this could be more clearly stated in the Abstract and the conclusions of the paper. The cocoon is the BLR around a less massive black hole, is this what the authors mean? But how would be the case where they have a BLR plus the cocoon, in objects B, D and -- to some extent also H? The cocoon surrounds the BLR?

- 2nd point I make: If the covering factor of this BLR/cocoon is close to unity, how does the ionizing radiation escapes to produce the narrow line component (from the NLR) present in all the spectra?

Referee #2

(Remarks to the Author)

I co-reviewed this manuscript with one of the reviewers who provided the listed reports.

Referee #4

(Remarks to the Author)

Referee Report

Referee expertise:

Referee #1 + 2: Multi-wavelength obs to study growth of SMBH

Referee #3: Space-based IR observations of obscured quasars

Referee #4: Galaxy-BH connection at high-z

Authors' note:

Thank you for all the insightful and detailed comments. We include our responses below respective comments. At the end we describe all changes that were not prompted by the referees' comments.

Referees' comments:

Referee #1 (Remarks to the Author):

Report on the paper "JWST's little red dots: an emerging population of young, low-mass AGN cocooned in dense ionized gas", by Rusakov et al.

The authors present an interesting analysis of "little red dots" (LRDs) - a class of high-redshift objects discovered by JWST that potentially harbor supermassive black holes (SMBHs). Using JWST NIRSpec observations of 13 sources, they argue that emission line profiles favor exponential shapes over Gaussian profiles, suggesting that electron scattering, rather than gas motion, is the primary mechanism for line broadening. They argue that this interpretation leads to considerably lower black hole mass estimates than previously reported. The authors propose that LRDs consist of low-mass SMBHs surrounded by obscuring Compton-thick gas that suppresses radio and X-ray emission, accreting at near-Eddington rates—effectively making them narrow-line Seyfert 1 analogs at high redshift. They argue that this model would help explain several puzzling aspects of recent LRD observations, including SMBHs that appear too massive for their cosmic age and their lack of expected multiwavelength counterparts.

While the authors present an interesting alternative scenario for interpreting LRDs, the physical interpretation require further justification as well as an improvement in the statistical evidence analysis before the conclusions can be considered robust. The authors should more thoroughly consider alternative explanations based on established AGN studies and provide clearer statistical evidence for their preferred model, answering to the aspects/concerns listed below.

1. Statistical Evidence for Exponential Profiles

The establishment of exponential line profiles over Gaussian profiles is crucial for the paper's subsequent interpretations. While I appreciate the detailed comparison in the Appendix (Figure 7 and Table 3), the main text lacks a clear statistical summary of model preference across the

entire sample. The BIC differences cited ($|\Delta\text{BIC}| > 10$) for high S/N objects do suggest evidence favoring the exponential or fiducial model, but a more comprehensive statistical statement is needed.

The authors should:

- Provide a clear summary of which model is preferred across the whole sample (e.g., "Model B was identified as the best model in X out of Y spectra")

This comparison is clarified on the lines 51-53.

- Transform ΔBIC values into model probabilities or for increased interpretability.

- Provide average weights across the dataset to quantify the overall statistical preference

We have calculated the weights of different models (as probabilities of individual model fits relative to the sum of the probabilities for different models) and we demonstrate them here by overplotting on our ED Figure 7 (please see below). As the decision for model rejection is made based on the values of ΔBIC (reported in Table 3) and so follows the broadly accepted approach and format in similar literature of using BIC or similar criteria (e.g., Harikane+23, Maiolino+2024, Juodzbali+2025, D'Eugenio+2025, Taylor+25), we would prefer to operate with the raw ΔBIC values. Such format may be helpful for reproducibility of these results and possible future comparisons in the literature.

2. Alternative Explanations Based on Known AGN Properties

The paper's interpretation conflicts with established knowledge of AGN emission line profiles in the nearby universe. Local AGN typically show: (i) A narrow component from low-density clouds (the Narrow-Line Region or NLR) with widths of a few hundred km/s extending to distances of a few hundred pc from the nucleus; (ii) A broad component from high-density clouds (the Broad-Line Region or BLR) with widths above ~ 2000 km/s at sub-pc distances.

Recent studies have shown that: (i) NLR profiles often require multiple Gaussian components, sometimes including a narrow and a broader component associated with outflows; (ii) BLR emission-line profiles are highly variable and rarely well-reproduced by a single Gaussian, often reflecting particular geometries (e.g., flattened rings) and kinematics involving outflows or inflows, a scenario confirmed by reverberation mapping studies.

Thus, although there may be the presence of electron scattering in the BLR, as argued by the authors, their test does not consider the possibility that the BLR may present a different profile than a Gaussian. They only compare the residuals of the fit of their model with that of a Gaussian.

While the AGN profile we find in these objects is unusual, as we point out, it is not wholly without precedent (e.g. Laor 2006). Indeed, it has been suggested that at low redshift high-accretion rate AGN may have more extended wings (Collin+2006). We now note this in the paper (line 210). We also note that we considered Lorentzian models that arise in turbulent gas in the inner BLR / outer accretion disk (Kollatschny+2013) or due to Raman scattering (Kokubo+2024).

As the Referee points out, several complex line shapes can be expected: e.g., double-peaked emitters at different systemic velocities due to eccentric disks or outflows (Eracleous+1995) or also top-flat profiles (Storchi-Bergmann+17). Biconical broad-line regions can produce either symmetric lines with logarithmic or gaussian wings or double-peaked emitters depending on the line of sight and exact geometry (Zheng+1990). Typically these shapes involve some kind of asymmetry or top-flat component which is not observed here.

Alternatively, a symmetric double-Gaussian BLR model which involves a very broad-line region (from outer accretion disk) and an intermediate broad-line region (from inner BLR) is possible. Therefore, we have extended our modelling to include an additional Gaussian component (i.e., a third Gaussian) and find that even these fits are worse than the fiducial/exponential in 8/13 fits or provide a similar goodness of fit in 5/13 fits albeit with more parameters. We include this and a further discussion in two new paragraphs in “Methods. Emission Lines”, and add this comparison in Extended Data Figure 7 and Table 3 (as “Double-Gaussian” model). A minor addition is made in the main text for context (on lines 49-50).

3. Interpretation of Line Components

A close examination of Figure 3 reveals patterns that may better align with conventional AGN interpretations:

- a) Their fits of the H α profiles of targets B and G are the only ones that present broad-line profiles (black line fits in Fig. 3) that have similar widths to those of the BLR of type 1 AGN in the near-Universe; the other 11 sources are argued to have very narrow BLR profiles (as shown by the black line fits);
- b) The authors argue that the profiles of the NLR are represented by only one narrow Gaussian (in blue in Fig. 3); but when I look at their fits, what they call the BLR component in the 11 sources (excluding B and G) has a similar or slightly larger width than that of the NLR, and could be identified with the second component of the NLR (as in the near-Universe), instead of arguing that it would come from the BLR;
- c) Inspection of the emission-line profiles of sources C, F, H, I, J and L show changes of curvature in the profile between the narrow core (from the NLR) and broad wings (from the BLR) that suggest that the wings are part of another component, as in the near-Universe: another set of clouds from the BLR that are denser, closer to the SMBH and move faster than those of the NLR.

The broad component we identify only has an existence as part of the convolution model (i.e. the components in black in Figure 3 convolved with corresponding exponentials). Indeed, the change in curvature indicates the presence of a narrow H-alpha and a broad H-alpha. The novelty here is the exponential shape of the broad component's wings (dominated by electron scattering) and its narrow Gaussian line core (from bulk gas motion) which are separate from the narrow H-alpha.

Adding an additional Gaussian component (i.e. using a double-Gaussian model for the broad component), as mentioned above typically gives a worse or equivalent fit to the data, again, indicating that these objects are not particularly well-described by a sum of Gaussians or that there is no preference for multiple Gaussians. We find that the highly symmetrical profiles militate against alternative scenarios involving complex kinematics or dynamics. Finally, in the double-Gaussian model used together with a narrow Gaussian component for the host galaxy we do not find evidence of additional NLR-like components with velocities < 1000 km/s. We comment on this alternative scenario in the MS in two new paragraphs (on lines 550 and 570, respectively).

We also include a discussion of possible narrow emission components arising from recombination in the electron scattering gas in and around the BLR (on line 181).

d) In line 49, the authors argue: "... relatively narrow Balmer absorption features are also a common feature in these objects"... But Fig. 3 shows this absorption in only on 3 of the 13 objects discussed, thus does not seem so common!

We intended common with respect to low-redshift AGN, where this feature is exceedingly rare. Such a feature is identified in only about a dozen AGN at low redshift, as pointed out in several previous works (e.g. in Zhang+2015; Schulze+2018), while is seen in at least 20% of the broad-line systems at high redshift (Matthee+24) and recent high-resolution observations (D'Eugenio+25) suggest they could be even more frequent in these high-redshift systems. We have now included a reference and modified the wording of this sentence to make this clear (lines 56-57).

e) In line 83, it is argued: : ... "The absence of strong broadening of the [O III] $\lambda\lambda$ 4959, 5007 doublet, while the H β line next to it is broadened, suggests that the electron volume density, n_e , in the scattering region, must be at least several times the critical density for these lines."... This is also the case in the usual interpretation of the BLR - origin of the broad Hbeta component, due to its higher density, supporting that the BLR and NLR in the LRDs present similar properties to those of nearby AGN.

f) The authors argue that the widths of the lines imply low black hole masses, on the assumption that the virial velocities is given by the widths of the lines. It is not clear if the authors are arguing that one should use the BLR width that they propose (the ones corresponding to the black profiles in Fig. 3 and listed under the profiles). These are too narrow; as pointed out above, most of them have widths typical of the NLR, while the virial relations use the actually measured FWHM of the broad profiles, not assuming a narrow profile and broadening it via electron scattering.

We revised the manuscript to clarify this point (on lines 75, 78, 142). We note here — the broadening of the observed broad line components for most of the sample is produced by electron scattering (>1000 km/s, as shown in Extended Data Table 4 and seen as black components in Figure 3), while the actual Doppler-broadened cores (<600 km/s, as deconvolved from the total broad profiles) are much narrower than previously inferred for these objects - implying lower BH masses. Thus, the intrinsic deconvolved BLR width (black profile) should be used.

g) Remark regarding Fig. 3: The labeling is partly difficult to read and interpret. For example, the meaning of "Gauss*Exp" is not immediately clear, neither that the black profile is that of the BLR component supposed to be scattered. I understood this only after reading the whole paper. Improved labeling and explanation (or perhaps separating some elements into additional figures) would enhance clarity.

We modified the legend in Fig. 3 to improve clarity.

4. Physical Plausibility

The authors propose that Compton-thick ionized gas surrounding the SMBH explains the lack of radio and X-ray emission in LRDs. However:

- The likelihood of Compton-thick obscuring structure being fully ionized at scales of ~ 100 light days requires further justification.

From the electron-scattered line widths we infer the spherical radii of a few light days for lowest expected densities in a broad-line region $n_e = 10^8 \text{ cm}^{-3}$, we clarify this on lines 98-99.

- Even with such obscuration, one might expect parsec-scale jets whose emission would not be obscured by the proposed material.

- The possibility that these objects are intrinsically radio-quiet should be more thoroughly considered.

This is certainly an interesting point. We now discuss the possibility of the suppression of jet formation (lines 159-160) and outflows with possible connections to gas metallicity in a new paragraph (line 218).

Referee #2 (Remarks to the Author):

I co-reviewed this manuscript with one of the reviewers who provided the listed reports.

Referee #3 (Remarks to the Author):

This paper presents a spectroscopic analysis of 12 broad-line galaxies (with $H\alpha$ FWHM > 1000 km/s), using high signal-to-noise (SNR > 5 near $H\alpha$) JWST/NIRSpec medium-resolution spectra. The dataset includes observations from CEERS, JADES, RUBIES, and 18 stacked spectra, spanning a redshift range of 3.4–6.7. For two of these galaxies, high-resolution spectra were also available. The authors argue that the observed line profiles are better fit by an exponential profile—consistent with Compton scattering in a dense medium—combined with a narrow Gaussian, rather than a single Gaussian profile typically associated with virialized gas motion around a supermassive black hole (SMBH). If correct, this result implies that previously reported black hole (BH) masses may be significantly overestimated and accretion rates underestimated, which would have substantial implications for models of SMBH formation and the interpretation of JWST-detected broad-line AGNs.

The paper proposes an intriguing and potentially important reinterpretation of broad-line profiles in JWST AGNs, raising fundamental questions about black hole mass estimates in the early universe. However, the manuscript requires major revisions before it can be considered for publication. The following concerns should be addressed:

Major Comments:

Alternative Line Profile Fits:

The authors dismiss double-Gaussian fits on the basis that such profiles would indicate dual AGNs, which they argue is unlikely for the full sample. However, broad wings in AGN line profiles have been previously reported (e.g., Nagao et al. 2006; Santos et al. 2025) and successfully modeled using double-Gaussian components. These features are often interpreted as arising from complex BLR dynamics, such as a combination of rotation and turbulence or distinct inner and outer BLR zones. The authors should revisit these alternative fits, incorporate a discussion of prior work, and evaluate how such interpretations might also lead to systematic overestimates in BH mass across AGN populations. Furthermore, they should justify why the exponential profile is favored in their data, but not in previous studies. A clear explanation of how the JWST broad-line AGNs differ—e.g., in terms of X-ray and radio properties—would strengthen the case for a distinct physical scenario.

In the response to referees 1&2 we include the double-Gaussian BLR model to our analysis (Extended Data Table 3, Figure 7). Two new paragraphs describe our findings in the Methods section (lines 549, 569). We also discuss objects with extended wings as part of two new paragraphs in the main text (on lines 205, 218).

We show that adding an additional Gaussian component still gives a worse fit than the exponential in most of our sample (especially in higher SNR spectra, except objects B, J) and an equal goodness-of-fit in the rest while having an additional parameter. Physically, we find that the wings are highly symmetric, disfavoring a wind/outflow/inflow interpretation for the components in emission, although Santos+2025 demonstrate similar symmetric double-Gaussian profiles in AGN at $z \sim 2$ which may be explained with stratified BLR or rotation+turbulence. We note that their work tests only the single and double-Gaussian profiles. Finally, our model comparison (Extended Data Table 3 and Figure 7) also excludes the basic Lorentzian shape compared to the basic Exponential, which rules out the turbulence as the dominant mechanism producing the broad line shape. We now comment on such alternative, more complex models in the Methods.

Indeed, multiple studies found broad lines with extended wings at lower redshift in Narrow Line Seyfert 1 (NLS1) galaxies, including, e.g. Veron-Cetty+2001 and Sulentic+2002 and noted that Lorentzian profiles produce a much better fit to the profiles. However, we are not aware of studies that included exponential profiles in their comparative analysis, except Laor+2006. We only note that Nagao+2006 found that power-law wings provide a better fit than the double Gaussians. Therefore, we comment in the manuscript that AGN with multi-Gaussian broad wings, especially the NLS1's, should be investigated for exponential profiles in the future. This would confirm the suggestion by Collin+2006 that higher Eddington-ratio objects tend to have more extended line wings. This discussion is included now in the new paragraph (on lines 205, 218) in the main text as well as in the new paragraph in Methods (lines 549, 569).

We make the case for the properties JWST BL-AGN in the first two paragraphs of the main text describing their general X-ray and radio properties. We then return to the issue of non-detections in the context of comparison with NLSy1's (line 205).

High-Ionization Lines:

The manuscript would benefit from a more thorough analysis of the rest of the spectrum, particularly with respect to high-ionization lines. The authors should comment on whether such lines are present or absent in their sample and, if absent, provide upper limits. A comparison with previously characterized JWST AGNs—some of which show such lines and others that do not—would add valuable context.

The presence or, in most cases, absence of high-ionization lines is particularly pertinent to the question of whether LRDs or broad line objects are AGN. We now comment on this for our sample and include references to other studies that investigate the presence of high-ionization lines in these and similar systems in the paragraph on line 40-44. On line 219 we also mention that we do not find evidence of Fe II emission.

Redshift-Dependent Cocoon Hypothesis:

The authors propose that dense, ionized gas cocoons are responsible for the exponential wings, but such structures are not observed in a significant population of lower-redshift AGNs (e.g., NLS1s). The paper should explore possible reasons for this redshift dependence. For instance, could metal-poor environments at high redshift play a role? Why are no local analogs found in similarly low-metallicity galaxies today? A more speculative discussion, while clearly labeled as such, would be appropriate here.

We include a new paragraph about a possible connection between our sample and the low-redshift narrow-line Seyfert 1 objects and their differences based on the presence of extended line wings (line 205). In the new paragraph that follows (line 218), we discuss more specifically the possible role of metallicity in reducing outflows and producing a smoother gas coverage in JWST objects and possible overlap or differences with the local AGN.

Comparison with Local NLS1s and High Accretion AGNs:

A more comprehensive comparison with narrow-line Seyfert 1 galaxies and other examples of high-Eddington accretors in the local universe would strengthen the paper. The authors should consider outflows seen in NLS1s and discuss why similar outflows in these high-redshift highly accreting systems may not be seen.

We have now considered this point in the paragraph on line 218. We include a note about the lack of outflows and radio jets in JWST objects as a likely distinction from some local NLS1s. We intend a full quantitative comparison to NLS1s in a future study.

Missing Intermediate Population:

The authors argue for a distinct population of JWST broad-line AGNs, but the lack of a transitional or intermediate population (with lower column densities) bridging the known high-redshift quasars and these new sources raises questions. Some discussion is needed on why such a population has not been detected and what this implies about AGN evolution, duty cycles, or observational biases.

This is an interesting question as well. We have now addressed this and the above two points (comparison with NLSy1, uniqueness to high redshifts and the evolutionary connections to other AGN types) in the new paragraphs in the main text (lines 205, 218).

Minor Comments:

Figure 2 should include a physical scale in kiloparsecs (kpc) for reference.
Implemented as suggested.

While case B recombination does not strictly apply at the gas densities inferred, it would be useful to provide the observed range of recombination line ratios and compare them to those seen in typical quasar populations.

While this is indeed a very important thing to do, it is a substantial additional analysis with many nuances. We therefore briefly discuss these ratios in the paper (line 187-190) and defer the full analysis to a future work (Nikopoulos et al. in prep.).

Referee #4 (Remarks to the Author):

The authors present a compelling list of arguments to interpret JWST's compact, distant galaxies showing broad lines. The H-alpha line broadening from scattered light instead of random motion in the broad line region is convincing, and so are the implications of Compton-thick ionized gas.

My main remaining concern is that the sample comes from a different color selection than typical LRDs (e.g., Extended Data Figure 2). The current work tries to explain LRDs with bluer galaxies than the LRDs. Could the authors check or comment on whether the scattered light interpretation for the broad H-alpha emitters could fully apply to the general populations of LRDs that appear redder on the color-color diagram? I would like to see at least a discussion on the potential impact (or the absence of it) of the sample selection. I assume if the majority of LRDs are missed by this work due to higher dust extinction, the line profiles could be intrinsically different. Do the spectra for A, E, H (also classified as LRDs) appear representative of the entire sample in this work?

We now comment in the new paragraph in the Methods (line 482) on the generalisability of inferences on LRD properties using A,E,H as examples, in particular, the Balmer nature of the spectra, the probable absence of high extinction (C,F,I) and the failure of the implemented colour selection criteria to efficiently select all LRD-like objects with an inflection point around the Balmer series limit and a diverse range of NIR slopes (C,D,G,H), as well as redshift effects combined with strong H β +[O III] lines. For e.g., the recent work (Hviding+2025) finds that the photometric criteria (such as the one employed in ED Fig. 2), have lower completeness than similar spectroscopic criteria, especially at bluer F277W-F444W colour and fainter rest-UV magnitudes. We note that the presence of election scattering and the magnitude of scattered broadening do not appear to depend on the classification.

Below are minor comments and questions.

Lines 119-120: Please correct me if I am wrong, trying to interpret the lines as a general reader: an outflow from star-formation requires 10^4 star clusters to produce 10^{45} erg/s of ionizing luminosity, but the gas masses (where do they come from, and how heavy are they?) require $>10^9$ solar luminosity per star cluster, requiring $>10^{13}$ solar luminosity for individual galaxies, ruling out the star-formation origin for the broad lines.

We clarified this detail in the text (line 126-127): the gas masses are based on the column and volume densities of free electrons. We calculate the gas masses of $\sim 10^5 M_{\text{sun}}$ for $n_e \sim 1e24 \text{ cm}^{-2}$ and $n_e \sim 1e8 \text{ cm}^{-3}$ (dense star formation above the critical density of [O III] such that its broadening is unobserved).

Lines 136-137: I thought the authors intended "consistent with the local BH-host galaxy scaling relation at lower redshifts".

This is now clarified.

Line 159 and lines 191-193: Is the spherical gas distribution a distinct feature at high-z, or in other words, are young AGN accretion seen only at high redshifts? Could the authors predict or identify any similar objects in the local universe, according to the formation mechanisms for LRDs the authors suggest?

We included the comparison with the narrow-line Seyfert 1 objects at low redshift, which are thought to be low-mass (10^5 - $10^7 M_{\text{sun}}$) AGN powered by high-Eddington accretion rate,

which makes them the most similar type of AGN to the LRDs as inferred here (new paragraph on line 218).

Line 165: Some LRDs are reported with high A_V values derived from the H-alpha to H-beta ratios (Balmer decrement). If the Balmer decrement of LRDs don't follow the case B values, how much could we trust the A_V values from the Balmer decrements?

We include this point to the list of predictions of our high-density gas model (on lines 188-190).

Extended Data Figure 6: I was curious if the trend became flat if the y-axis was replaced by the difference in chi-squares divided by the chi-squares of the exponential model, or replaced by the ratio of the chi-square values. If so, it would imply the exponential model fits better than the H-alpha profile at any S/N. I recall the F-test where the difference in the chi-squares divided by the reduced chi-squares of the exponential model should follow an F-distribution.

Yes, the exponential gives a good fit at any SNR. However, what this figure shows is that the Gaussian model becomes increasingly worse at higher SNR. Therefore, the suggested plot $\Delta \chi^2 / \chi^2$ vs SNR still shows an inclined trend with SNR.

We note the Gauss and Exp models have the same number of degrees of freedom (degrees of freedom are shown in Extended Data Table 3), but for the gaussian and fiducial model comparison, the F-test indeed gives p-values of $p \sim 10^{-4} - 10^{-16}$ (except $p=1$ for object G), i.e. strongly suggesting the improved fit with the fiducial model. We note that the delta BIC test tends to be stricter with more parameters and larger data sizes and therefore we use the delta BIC comparison as the more conservative one.

Extended Data Figure 7: Why are in some cases, the fiducial model has a poorer fit than the exponential? I guess the added degrees of freedom for the fiducial model requires a significantly improved goodness-of-fit for a larger Delta BIC value. Also, does the scaling of the reduced chi-square values to 1.4 change the Delta BIC values?

Indeed, the two additional parameters in the fiducial model moves this model further from the basic exponential by about $\Delta \text{BIC} = 2 * \log_2(>100-200) = 10-15$ for equal values of chi-square. This can be seen particularly for objects A and E. For A, the exponential and fiducial shapes are almost equivalent, likely because the narrow core is very narrow compared to the spectral resolution. While for object E, the line core is hard to constrain as it is almost entirely absorbed, while the wings are clearly exponential.

The scaling by 1.4 does change the BIC values – the chi-squares are divided by 1.4, but the penalty term in $\text{BIC} = \chi^2 + k * \log(n)$ is unchanged. Our best model choices before and after this step do not change (we now note this in the caption of the figure).

=== Further notes from Authors =====

Here, we would like to note all changes/additions that we made that were not included in the responses above, which include comments based on the recently published literature.

One name is added to the author list (Joris Witstok). Their name was not added at the beginning due to a mistake in managing the lists of authors from two main groups involved in this work.

Throughout this work, we introduce a minor change in the naming of the objects in our sample. Instead of "FIELD Program ID-Source ID" (e.g., GN 1181-68797), we are now using "NAME(or Program ID)-Field-Source ID" (e.g., JADES-GN-68797) (see Extended Data Table 2). This is

done to make sure we are using the same naming convention as usually done in papers using the RUBIES and JADES surveys' data.

Previously, during our model fitting Object D was fitted with a simplified fiducial model which excluded a non-scattered broad-line Gaussian which is included in all other object fits with the fiducial model. This was done because otherwise the presence of a relatively low-SNR P Cygni feature (described by four parameters alone) made the optimization more complex and resulted in poorly constrained posterior distributions with MCMC. Now, we revised this fit by including the non-scattered Gaussian and increasing the number of MCMC walkers to constrain the solution. As a result, the width of the intrinsic BLR Gaussian has changed from FWHM~300 km/s to FWHM~1500 km/s - one of the broadest intrinsic Gaussians in our sample, although still dominated by the electron scattering broadening (FWHM~1600 km/s from scattering). We noted this on lines 79-81. The values for this object are changed accordingly in Figures 3, 4 Table 1, Extended Data Table 3, 4 and Extended Data Figure 7.

Previously, Table 3 showed the chi-square and BIC values without scaling the chi-square values by the reduced chi-square factor of 1.4 (to account for possibly underestimated uncertainties in the spectra), while corrected BIC values were shown in Extended Data Figure 7. We fix this and report the corrected values in ED Table 3 as well (and include a note in the caption).

Figure 3 caption clarifies the limits on the range of FWHM posterior values in the insets.

Figure 4 caption includes a comment with a reference regarding the stellar mass estimates.

On lines 170-174 we make a note about the high degree of symmetry of our broad lines and that this likely suggests a lack of strong outflows and that most of the broadening is likely not produced in the accretion disk.

On lines 180-187 we add a prediction for the geometry and location of the scattering medium in these objects based on their optical depth and predominantly exponential shape of the wings, inspired by a comment regarding this in Juodzbališ+2025.

Extended Data Table 2 includes a reference to object E being previously reported in the literature.

Corner plots in Extended Data Figures 9 and 12 previously included an incorrect axis label for one of the parameters: log A (H-alpha broad non-scat.) instead of A (H-alpha broad non-scat.). This is now corrected.

Very minor text changes to improve clarity: lines 59-60, 75, 87-89, 92-93, 140, 189, 203-204, 506-507, 519-520.

Referee Report

Authors note:

Thank you to all referees for the second review report. We include our responses to the comments below. At the end we describe all changes we did in addition to the referees' comments.

Referees' comments:

Referee #1 (Remarks to the Author):

The authors answered most of the questions that we have raised. We appreciated, in particular the inclusion of the model using two Gaussian components for the BLR that they show do improve the fits, showing similar fitting goodness as their models in 5/13 of the sources. But in the others their model gave better fits. The fact that the wings show preferably exponential profiles is also in favor of their models.

We conclude that their proposed model applies to many emission-line profiles of the LRDs, but not necessarily for all the sources. In particular:

(i) Sources B, D and G (3/12, thus 25%) show H α broad (without electron broadening) that are broader than 1500 km/s, which support the presence of a SMBH with mass $\sim 10^8$ solar masses; thus, in these cases, the width of the lines are dominated by the SMBH gravitational potential;

(iii) The number of sources with absorption is small (3/12, the same as the number of cases cited above in (i) that support the presence of a massive SMBH);

(ii) Sources I, J, K, L (4/12) and the Stack S in Fig. 3 seem to show H α broad component narrower than the narrow component due to the host galaxy. Does this make sense?

We note that the intrinsic Gaussian BLR H α components of systems J, K and the stack S are broader than the narrow H α component or consistent with it, while objects I, L have solutions that are 2-sigma upper limits which are also consistent with the narrow Gaussians, but may be narrower. We show these Gaussian FWHM values below.

Object	BLR Gaussian / km s ⁽⁻¹⁾	Narrow Host Gaussian / km s ⁽⁻¹⁾
I	<274	267 + 24 - 34
J	197 + 42 - 56	139 + 9 - 12
K	311 + 80 - 70	147 + 11 - 15

L	< 155 or 627 + 203 - 122	167 + 63 - 54
Stack	339 + 55 - 51	226 + 19 - 21

The observed narrow H-alpha is likely made of several components that are decoupled from the dominant dynamical influence of the black hole: (i) the host galaxy emission and (ii) narrow line region components. Also, we expect the H-alpha to include a component coupled with the BLR of the black hole dynamically: a component formed by recombination of the electron-scattering gas. As the scattering medium is likely in or around the BLR, we expect its width to be the same as in the Gaussian BLR component. Therefore, it may be expected that the BLR-related components may be narrower than the other H-alpha components if the black hole has a correspondingly lower mass.

We note that our sample is in general consistent within 1-2 sigma with the $M_{\text{BH}} - \sigma$ relation (e.g., from Bennert+2021), but objects IJKL are more than 2 sigma below the relation indeed. One could expect such deviation if the locally calibrated scaling relation doesn't apply to these systems or if the scatter in the relation is greater. We note this observation in **Methods on lines 616-639 (in a new paragraph on the reliability of our black hole mass estimation)**.

(iv) Fig. 3 shows that the broad profile of several sources are somewhat asymmetric relative to the center wavelength ($H\alpha_{\text{Narrow}}$): we see some asymmetries (in decreasing order) in H, L, I, F, C and J.

Indeed, the best-fit centroid position of the BLR is slightly offset from the narrow H-alpha in several systems. However, the offset is either not significant or within the spectral resolution. We show those values here for reference:

Object	Offset / km s^{-1}	FWHM resolution / km s^{-1}
C	-20 +/- 4	77
F	104 +/- 17	193
H	131 +/- 23	177
I	-31 +/- 31	238
J	149 +/- 14	154
L	-31 +/- 26	168

In summary, we do agree that the presence of electron scattering is important to describe the wings of the line profiles and that their model is a good alternative to explain many LRDs, but not necessarily all.

We add a note **on lines 204-207** that the electron-scattering model doesn't apply to all our examples, such as objects B and G, and that electron scattering accounts for a fraction of broadening in object D, as opposed to completely dominating the width in other cases.

Referee #2 (Remarks to the Author):

I co-reviewed this manuscript with one of the reviewers who provided the listed reports.

Referee #3 (Remarks to the Author):

I thank the authors for carefully addressing my comments. My comments have been adequately addressed to warrant publications with only one minor point. In the new paragraph (line 218) addressing my question about local objects, the authors should also mention the growing population of metal poor compact dwarfs that show broad lines and [NeV] emission despite being metal poor. Recent JWST observations are uncovering more of such objects where the AGN origin of the broad lines is being questioned. Some comment on this population with references to the papers (e.g. Izotov et al. 2021, Reefe et al. 2023, Hatano et al, Mingozi et al. 2025) should be mentioned.

Thank you for these references – these are intriguing observations. It appears that several of these papers investigate IMBH scenarios indirectly via photoionization modelling (Izotov+21, Hatano+24a, Mingozi+25), and others find evidence of broad lines with non-Gaussian extended wings (Reefe+23, Hatano+24b). We add latter references #66, 67 as most suitable to our discussion **on line 222**, and leave the other for now as we face the limit on the number of references in the main text.

Referee #4 (Remarks to the Author):

I now better see (correct me if I am wrong) that the fiducial model profile to fit the H-alpha lines consists of a non-scattered and a scattered (the same non-scattered Gaussian but convolved with an exponential kernel) component of an intrinsically Doppler-broadened Gaussian, and that the authors use the intrinsic Gaussian width to estimate the BH mass. Although the statistical reasoning to adopt such scattering over pure Doppler broadening appears sound, the implications of the Doppler-broadened line widths is worrisome. From the inset of Fig 3 and Table 4, the intrinsic line widths appear mostly narrower than 600 km/s, and comparable to the broadening at the narrow-line region. Moreover, objects F, I, J, K, L, S have the intrinsic Gaussian as narrow as their narrow lines, further supporting that the H-alpha line may not be originating from the broad-line region (hereafter BLR). AGN virial BH mass estimators rely on the Doppler broadening of the BLR, defined to be about 1000 km/s or broader. Could the Doppler-broadened intrinsic H-alpha profiles from the targets, be interpreted as coming from the BLR? I would rather believe that most of the targets don't have a BLR, which is a radical idea scarcely seen in existing literature on AGN evolution.

I tried to search from the literature if there are existing discussions on intermediate-line (FWHM<1000 km/s but broader than narrow lines) regions that might be responsible for

explaining million-solar-mass BHs the authors try to in Fig 4 (e.g., Hu et al. 2008, ApJL, 683, 115), but couldn't find emission lines much narrower than 1000 km/s used or interpreted as broad or even intermediate. Bontà et al. (2020, ApJ, 903, 112) use reverberation mapped H-beta calibrations down to 800 km/s but not any narrower (and likewise for the low-L studies by Bentz et al. 2009, ApJ, 705, 199, and later work by their group). The apparent absence of a typically broad-line is predicted (Elitzur & Ho 2009, ApJL, 701, 91) at bolometric luminosities of around 10^{39} erg/s and below, which is much lower than the 10^{43} - 10^{45} erg/s found in this work (assuming the H-alpha is a few percent of the bolometric, line 198). All of my inquiries lead to failure in explaining the narrow line profiles of the targets.

I thus request the authors to consider the implications of the potential absence of a BLR: perhaps i) the AGN is in its earliest evolutionary phase of forming a BLR (but then I don't think the BH mass estimators could be used as the current way: perhaps an upper limit on the line widths of 1000 km/s should be imposed instead for targets with $\text{FWHM} < 1000$ km/s), or ii) we are looking at very strange/rare AGN without the BLR, iii) the LRDs are not AGNs and the lines more likely come from star-formation, or iv) the scattering interpretation has a missing piece of the puzzle requiring further investigation.

Thank you for providing such detailed information. It is indeed an interesting possibility that the intrinsic Gaussian components have a non-BLR origin.

We originally explored the scenario (iii) and devoted a paragraph for this discussion **on lines 115-136**. As the luminosity density of $1e45$ erg/s/pc³ appears to be four orders of magnitude greater than a highest-density known star-forming system, we concluded that accretion is the most likely mechanism to produce such luminosity in a volume given by the size of the electron scattering medium (order of magnitude ~ 1 pc).

Comment on scenario (ii): It may be expected that AGN in their early evolutionary stage have a lower mass black hole, which sets the dynamics of the BLR accordingly and results in relatively narrow < 1000 km/s bulk velocity. Although intrinsically narrow, this H-alpha component is distinct from the narrow lines in the spectra and therefore there has to be a BLR gas to produce these lines. Although of course this region is intrinsically much narrower than that of typical Type I AGN or quasars, it appears to be "broad" after being scattered in the electron gas in or immediately outside the BLR. The reference to Elitzur & Ho 2009, ApJL, 701, 91 is very interesting – thank you. It appears that their scenario for BLR disappearance is driven by the low disk-wind column densities in low-luminosity AGN with very low accretion rates – therefore such scenario would not be expected in our case, as the accretion rate appears to be high and the lines are clearly detected.

In response to point (i) – indeed, these systems appear to be unique and likely in their early evolutionary stage, as suggested by the narrow Doppler width and a unique combination of other properties of these objects (compactness, red continuum, X-ray and radio non-detections and likely low metallicity).

We agree that the BH mass estimator used here or others in the literature appear to have not been either calibrated or tested in the regime of narrow Doppler widths ($\text{FWHM} \ll 1000$ km/s). One likely distinction of the BLR of LRDs from those used to define the estimators is the BLR covering fraction. We use an estimator from Reins & Volonteri (2015) which assumes $M_{\text{BH}} \sim L(\text{Ha})^{0.5}$. If LRDs have a complete covering fraction of 1.0, compared to typically assumed factors of 0.1-0.5, their current masses may be overestimated by up to a factor of ~ 3

($10^{0.5}=3.1$). This is comparable to a scatter of 0.5 dex of single-epoch estimator masses (Shen+2013). We caution about these and other effects in the caption to Figure 4 and in a new Methods section **on lines 616-639 “Reliability of the black-hole mass estimation”**.

(Authors): Finally, below we list the changes we made in addition to those prompted by the referees:

- The title is shortened to 77 characters to match as closely as possible the guidelines: “Little red dots: young supermassive black holes cocooned in dense ionized gas”
- We add a caution (at the end of Figure 4 caption) that the black hole masses in our work may be underestimated if the BLR H-alpha component is affected by dust extinction. This will be studied in a future work.

We remind here that we avoid inferring dust extinction from Balmer line decrements because the Balmer lines may be optically thick and the recombination physics may deviate from the standard Case B scenario, in which case the Balmer decrements cannot be used as a dust extinction diagnostic (**lines 187-189**).

- Figure 4 caption is shortened.

Response to Referee Report (Manuscript 2025-03-06991B)

Authors note:

Thank you to Referees 1 & 2 for the latest report. We include our responses to the comments below and make changes in the manuscript in bold font.

Vadim Rusakov (on behalf of all Authors)

Referees' comments:

Referee #1 (Remarks to the Author):

The paper has improved, and their model provides good fits to the profiles of most of the data. But I still see a few inconsistencies and suggest small modifications to take them into account, as follows.

- In the Abstract, the authors write: "Using the highest-quality JWST spectra, we show here that the lines are broadened by electron scattering with a narrow intrinsic line core."
- To be in agreement with their findings they should add: "in 9/12 (or in most) of the studied sources", because in three sources the intrinsic line core is not narrow, but a broad component is necessary, indicating more massive black holes. Thus, at least in 25% percent of the sources, there is a massive enough black hole to broaden the line.

We now use the above suggestion and clarify that this is found "in most objects studied ...".

- Caption of Fig. 3: ..." in several cases, A P Cygni profile or a Gaussian Absorption feature are included".

- I see this only in 3/12 of the profiles, a minority, thus "several" is misleading; they should state "in 25%" (or "in some") of the sources".

We now state explicitly "In objects A D E, ...".

- Regarding the physical scenario: The authors argue for the presence of an accreting black hole that has emission from the accretion disk cocooned in dense ionized gas. Still, there is escaping radiation to generate the NLR emission. E.g., in line 625, they write: "Furthermore, objects in our sample likely have a BLR covering factor close to unity (i.e. the fraction of the solid angle of a sphere, $\Omega/4\pi$) compared to the typically assumed values 0.1 – 0.5, as that would help to explain the weakness of X-ray and radio emission that occurs ubiquitously (see Methods below and the main text). Thus, if the factor is ten times greater,

the masses here may be overestimated by up to a factor of ~ 3 , as they scale as $M_{\text{BH}} \propto L(\text{H}\alpha)^{0.5}$

- 1st point I would like to make: they are calling the cocoon the BLR? Maybe this could be more clearly stated in the Abstract and the conclusions of the paper. The cocoon is the BLR around a less massive black hole, is this what the authors mean? But how would be the case where they have a BLR plus the cocoon, in objects B, D and -- to some extent also H? The cocoon surrounds the BLR?

The cocoon includes the BLR-like rotating gas (we don't call it explicitly BLR to avoid implying large rotation velocities), the scattering ionized gas (which likely overlaps with the BLR gas, based on the absence of strong narrow H α component from recombination in the ionized scattering medium suggested by Juodzbališ et al. 2024 arXiv:2504.03551), the gas in the $n \geq 2$ state in very mild inflows, outflows (which sometimes produce detectable absorption features; discussion on lines 85-92). In the previous revision, we devoted part of our discussion to details clarifying the suggested picture (on lines 137-144).

We would like to defer the detailed kinematic and ionization structure, as well as geometry to a future study that performs a devoted radiative transfer analysis, to avoid strong speculations. In abstract, we add more details to the cocoon description: "They are enshrouded in a dense cocoon of ionized gas from which they are accreting close to the Eddington limit, **with very mild neutral outflows**".

- 2nd point I make: If the covering factor of this BLR/cocoon is close to unity, how does the ionizing radiation escapes to produce the narrow line component (from the NLR) present in all the spectra?

We note the observed narrow line emission likely comes (at least in part) from star forming regions in the host galaxy (in caption of Figure 3 which shows the fiducial model; and on lines 382-384 when describing our line model), while the cocoon is expected to reprocess most of the ionizing radiation, likely without producing strong NLR emission.

Referee #2 (Remarks to the Author):

I co-reviewed this manuscript with one of the reviewers who provided the listed reports.